# Estimation of 1 km downwelling shortwave radiation over the Tibetan Plateau under all-sky conditions

Peizhen Li[1], Lei Zhong[1,2,3,4], Yaoming Ma[5,6,7,8,9,10], Yunfei Fu[1], Meilin Cheng[1], Xian Wang[1], Yuting Qi[1], Zixin Wang[1]

[1]School of Earth and Space Sciences, University of Science and Technology of China, Hefei 230026, China

[2]CAS Center for Excellence in Comparative Planetology, Hefei 230026, China

[3]Jiangsu Collaborative Innovation Center for Climate Change, Nanjing 210023, China

[4]Frontiers Science Center for Planetary Exploration and Emerging Technologies, University of Science and Technology of China, Hefei 230026, China

[5]Land-Atmosphere Interaction and its Climatic Effects Group, State Key Laboratory of Tibetan Plateau Earth System, Resources and Environment (TPESRE), Institute of Tibetan Plateau Research, Chinese Academy of Sciences, Beijing 100101, China

[6]College of Earth and Planetary Sciences, University of Chinese Academy of Sciences, Beijing 100049, China

[7]College of Atmospheric Science, Lanzhou University, Lanzhou 730000, China

[8]National Observation and Research Station for Qomolongma Special Atmospheric Processes and Environmental Changes, Dingri 858200, China

[9]Kathmandu Center of Research and Education, Chinese Academy of Sciences, Beijing 100101, China

[10]China-Pakistan Joint Research Center on Earth Sciences, Chinese Academy of Sciences, Islamabad 45320, Pakistan

*Correspondence to*: Lei Zhong (zhonglei@ustc.edu.cn)

**Abstract.** Downwelling shortwave radiation (DSR) is the basic driving force for the energy and water cycles of the Earth's climate system. As called the Third Pole of the Earth, the Tibetan Plateau (TP) absorbs a large amount of shortwave radiation and exert important impacts on global weather and climate change. However, due to coarse spatial resolution and insufficient consideration of factors influencing radiative transfer processes, DSR parameterization schemes are still need to be improved when applied to the TP. Based on satellite datasets and meteorological forcing data, all-sky DSR over the TP at a spatial resolution of 1 km was derived using an improved parameterization scheme. The influence of topography and different radiative attenuations were comprehensively taken into account. Specifically, the introduction of cloud multiscattering and topography factors further improves the DSR estimation accuracy. The validation results indicated that the developed parameterization scheme showed reasonable accuracy. By comparing with current widely used DSR products based on the same in situ observations, the derived DSR performed much better on different spatial and temporal scales. On

instantaneous, ten-day, and monthly timescales, the root-mean-square errors (RMSEs) of the derived

DSR are 132.8~158.2 W m$^{-2}$, 70.8~76.5 W m$^{-2}$, and 61.3~67.5 W m$^{-2}$, respectively, which are much

smaller than those of current DSR products. The derived DSR not only captured the temporal variation

characteristics that are more consistent with the in situ measurements, but also provided reasonable

spatial patterns. Meanwhile, the proposed parameterization scheme demonstrated its superiority in

characterizing more details and high dynamics of the spatial pattern of DSR due to its terrain correction

and high resolution. Moreover, this parameterization scheme does not need any local correction in

advance and it has the potential to be extended to other regions in the world.

**1 Introduction**

Solar radiation is the basic energy source for surface biological, physical and chemical processes

(vegetation photosynthesis, evapotranspiration, plant and crop growth, etc.) (Wang et al., 2015; Liang

et al., 2019). It plays an important role in surface energy balance, land–atmosphere interactions,

weather and climate change (Li et al., 1997; Wang and Dickinson, 2013; Huang et al., 2019).

Furthermore, it is the key input data for land surface process models, hydrological models and earth

system models (Pinker et al., 2005; Liang et al., 2010; Stephens et al., 2012; Letu et al., 2020).

The Tibetan Plateau (TP) covers an area of approximately 2.65 million square kilometers. It is

known as the "Roof of the World" and "the Third Pole of the Earth" because of its average altitude of

more than 4000 m (approximately 1/3 of the troposphere height) and extremely complex topography

(Qiu, 2008; Yao et al., 2012). In addition, the TP and its surrounding areas hold the largest number of

glaciers outside the polar regions (Yao et al., 2012). The Yangtze River, the Yellow River, the Indus

River and most major rivers in Asia originate from the TP, and thereby the TP is also called the "Asian

Water Tower" (Xu et al., 2008; Immerzeel et al., 2010). Therefore, the unique features of the TP make it

an important research object for global and regional energy and water circulation and is one of the most

sensitive regions in response to global climate and environmental change.

Due to its high altitude, low airmass and short path for solar radiation to reach its surface, the TP

receives a large amount of radiation (Yang et al., 2014; Ma et al., 2017). The analysis of existing

observation data shows that the solar radiation heating effect of the TP is obviously stronger than that

of surrounding areas. Even the measured downwelling shortwave radiation (DSR) exceeds the solar

constant that occurs frequently (Tanaka et al., 2001; Yang et al., 2006b; Yang et al., 2008). As a result,

the TP can generate an intense surface heating field, which drives atmospheric circulation, regulates the

formation and development of the East Asian monsoon, and exerts an important impact on global

weather processes and climate change (Hong et al., 2012; Wu et al., 2012; Zhao et al., 2018; Zhao et al.,

2019b). Radiation-related changes to the environment become more severe in the case of global

warming, such as significant snow melt, glacier retreat and permafrost thawing (Piao et al., 2010; Yang

et al., 2010b; Kuang and Jiao, 2016). In turn, these processes may pose a threat to engineering

constructions such as the Qinghai-Tibetan highway and railway (Chen et al., 2006; Yang et al., 2010a).

Meanwhile, in the context of carbon neutrality, DSR has become not only a vital source of energy for

local residents, but also an indispensable part of photovoltaic energy technologies (Zhang et al., 2017;

Huang et al., 2022; Yang et al., 2022). Consequently, reliable DSR estimation over the TP is of great

value for many studies and related practical applications.

    For many years, in situ measurements, numerical modeling, and satellite remote sensing have been

three effective ways to obtain DSR (Liang et al., 2019). In situ measurements are the most direct and

reliable way to obtain DSR data with high accuracy and high temporal resolution. However, due to the

high maintenance cost of field instruments, DSR observations are available at a smaller number of

stations compared to other routine meteorological variables, such as air temperature, pressure and

humidity, especially in areas with harsh climate conditions (e.g., Antarctica, the Arctic and the TP). In

situ measurements of DSR in these regions are not only sparse but also unevenly distributed. It is

therefore not enough to characterize the distribution pattern of DSR at a large spatial scale. Numerical

models can provide spatiotemporally continuous DSR data at regional and global scales. However, the

spatial resolution is relatively coarse (Decker et al., 2012). The accuracy is limited due to the

uncertainties of models in simulating or predicting cloud quantities. In contrast, satellite remote sensing

technology has certain advantages in estimating DSR with high spatial resolution over a large spatial

coverage. The sensors aboard satellites can dynamically monitor the evolution and spatial distribution

of clouds and capture a large amount of information about the atmosphere and underlying surface.

    During the past few decades, various satellite-based methods for estimating DSR have been

developed, which can be roughly divided into two categories: statistical methods and methods based on

radiative transfer processes (Sengupta et al., 2018; Huang et al., 2019; Letu et al., 2020). The statistical

methods used to estimate DSR construct the functional relationship between satellite measurements and in situ observations. Traditional empirical methods are simple to operate by applying statistical regression (Masuda et al., 1995; Li et al., 1997). However, the empirical model may work at the local

scale but needs recalibration over different regions. Artificial intelligence models, which can estimate DSR by building nonlinear relationships between satellite signals and ground-based DSR, have become a new trend to estimate radiation flux (Lu et al., 2011; Qin et al., 2011; Wei et al., 2019; Ma et al., 2020a). However, owing to an insufficient physical basis, the calculation accuracy of such methods depends largely on the selection of training data, and consequently, their generalizability is limited. In

addition, the artificial intelligence model usually needs a large number of samples to train the model. Therefore, due to insufficient ground-based observations, this method is not easy to apply in the TP (Yang et al., 2010a; Zhang et al., 2015). The look-up table (LUT) and physical parameterization method (Pinker and Laszlo, 1992; Bisht et al., 2005; Liang et al., 2006; Lu et al., 2010; Xie et al., 2016; Tang et al., 2019) are two typical methods based on the radiation transfer process and have been widely

used to estimate DSR from satellite data. Since LUT is a close approximation to the complicated radiative transfer model (RTM), a large number of parameters are needed as inputs, such as cloud, aerosol and atmospheric parameters, to obtain higher estimation accuracy. However, the data volume in the LUT will be greatly increased, which will further reduce the estimation efficiency of DSR. At the same time, it is usually necessary to encrypt the discrete calculation results by means of complex

interpolation algorithms (Letu et al., 2020), which in turn will lead to a computational load. Alternatively, the physical parameterization method can alleviate the computational burden by parameterizing the complex process in RTM while maintaining sufficient estimation accuracy.

      To date, the DSR parameterization scheme under clear-sky conditions has been quite mature (Bisht et al., 2005; Gueymard, 2012; Hwang et al., 2012). However, since optical remote sensing is

greatly affected by clouds, it is still a problem to be solved to estimate DSR efficiently and accurately under all-sky conditions (Li et al., 1995; Li et al., 1997; Huang et al., 2019; Zhong et al., 2019; Letu et al., 2020). Although some studies have proposed parameterization schemes for cloudy-sky conditions, the current schemes still have some defects. In the presence of clouds, cloud-sky parameterization, which only considers cloud fraction and cloud optical thickness, is usually coupled into clear-sky

models in a simple and arbitrary manner (Niemela et al., 2001; Bisht and Bras, 2010). Second, some

parameterization schemes did not consider the DSR attenuation caused by clouds carefully enough. Generally, only the single scattering of clouds was considered, and the multiple scattering effect of clouds was ignored (Huang et al., 2018; Huang et al., 2020).

Due to the high elevation and complex terrain of the TP, the impact of terrain on DSR should be taken into account. Tovar et al. (1995) found that there is no obvious correlation between the spatial variation in radiation in mountainous areas and interstation distance, but it varies with the altitude difference. Therefore, the DSR in mountainous areas cannot be estimated simply by interpolation of adjacent observation values. Yang et al. (2006b) pointed out that GEWEX-SRB v2.5 greatly underestimated the DSR on the TP due to ignoring the influence of surface elevation. Olson and Rupper (2019) reported that the deviation of the surface radiation budget could exceed 40 W m$^{-2}$ during the summer melting season in the high-mountain Asia area. In addition, the coarse spatial resolution of most existing DSR products is prone to cause uncertainties in rugged areas such as the TP. Currently, the spatial resolution and accuracy of most existing DSR products cannot meet the requirements of energy and water cycle studies over the TP (Zhong et al., 2019a; Wang et al., 2021; Zhang et al., 2022). Therefore, all-sky DSR products with fine spatial resolution and high accuracy over the entire TP are still lacking.

In general, some existing DSR estimation methods are still not applicable to the TP due to its highly variable terrain, high elevation, and unique climatic conditions. Therefore, an effective method to estimate the DSR of the entire TP under all-sky conditions is urgently needed. In this study, an improved parameterization scheme is proposed, and the derived DSR is validated by in situ measurements and compared with various existing DSR products. Then, the spatiotemporal distribution of the estimated DSR is presented, and the improvement brought by considering the multiple scattering effect of clouds and topographic factors is discussed. The paper is organized as follows: Section 2 introduces the input data and validation data. Section 3 introduces the improved parameterization method. Section 4 presents the results and discussion. The main conclusions are given in Section 5.

**2 Data**

**2.1 Input data**

The basic information of the meteorological forcing data and satellite datasets are listed in Table 1. The

China Meteorological Forcing Dataset (CMFD) has a temporal resolution of 3 hours and a horizontal spatial resolution of 0.1°. It has been widely used by the scientific community due to its high resolution and quality. These forcing data were produced by combining routine meteorological observations of the China Meteorological Administration (CMA), Princeton reanalysis datasets, the Global Land Data Assimilation System (GLDAS), the GEWEX Surface Radiation Budget (GEWEX-SRB) and the Tropical Rainfall Measuring Mission (TRMM) satellite (He et al., 2020). The surface air pressure (Pa), air temperature (K) and specific humidity (kg kg$^{-1}$) are used for DSR estimation.

The satellite data come from MODIS, OMI and ASTER. The inputs for the parameterization scheme include (1) the cloud phase, cloud water path (CWP), cloud effective radius (CER) (MODIS cloud product MOD06_L2), (2) aerosol optical depth (AOD) (MODIS aerosol products MOD04_L2), (3) ground surface albedo (MODIS Combined Land Albedo Product MCD43C3), (4) geolocation information (MOD03), (5) total ozone column amount (OMTO3e), and (6) 30-m ASTER digital elevation model.

The MODIS combined Dark Target and Deep Blue AOD at 0.55 μm for land and ocean were used to derive the aerosol Ångström turbidity coefficients (Kim, 2004; Yang et al., 2006a; Huang et al., 2018). The actual surface albedo is derived with the shortwave black sky albedo (BSA) and white sky albedo (WSA) from the albedo product (Schaaf et al., 2002; Pinty et al., 2005). All MODIS product versions mentioned above are in collection 6. The OMI science team created the OMTO3e product by selecting the best pixel data from the high-quality filtered level-2 total column ozone data (Ahn et al., 2008).

It should be noted that in operational applications, many parameters may not be available, especially in areas with extreme climatic conditions, such as the TP. Therefore, the "gap-filling" procedure should be carried out first, as in most studies. For aerosols, the invalid retrievals would be substituted using the Level-3 MODIS global daily and monthly climatological products (Qin et al., 2015; Huang et al., 2016a; Li et al., 2022). For the ozone column amount and surface albedo, the unavailable retrievals were substituted using the nearest valid retrievals (Huang et al., 2018; Tang et al., 2019; Zhong et al., 2019b). The spatial resolutions of MODIS aerosol and albedo data are 10 km and 5 km, respectively. The spatial resolutions of ozone and DEM data are 25 km and 30 m, respectively. To obtain the DSR at the 1 km spatial scale, these data were resampled to 1 km.

**Table 1.** Overview of the meteorological forcing and satellite datasets used in this study.

| Data sources | Product name | Variable | Spatial resolution | Temporal extent |
|---|---|---|---|---|
| CMFD | - | Temperature<br>Pressure<br>Specific humidity | 0.1° x 0.1° | 1979 to 2018 |
| MODIS | MOD06_L2 | Cloud phase<br>Cloud water path<br>Cloud effective radius | 1 km | 2000 to present |
| | MOD04_L2 | Aerosol optical depth | 10 km | |
| | MCD43C3 | Black-Sky albedo<br>White-Sky albedo | 5 km | |
| | MOD03 | Latitude<br>Longitude<br>Solar zenith | 1 km | |
| ASTER | AST14DEM | DEM | 30 m | 2000 to present |
| OMI | OMTO3e | Total column ozone | 0.25° x 0.25° | 2004 to present |

**2.2 In situ observation stations**

The distributions of the in situ observation stations are marked in Fig. 1, and their basic information is listed in Table 2. In this study, in situ DSR observations used to validate the accuracy of the improved parameterization scheme were extracted from 12 stations over the TP. A variety of elevations, climates, and land cover types are included in these validation stations. Among them, six stations are obtained

from the Tibetan Observation and Research Platform (TORP) project (Ma et al., 2008), including BJ, QOMS, SETORS, NADORS, MAWORS and NAMORS stations. These six stations composed an integrated high-elevation and cold-region observation network. More detailed descriptions of these six stations are described by Ma et al. (2020b). The Xidatan (XDT) monitoring station representing the characteristics of discontinuous and warm permafrost is located along the northern permafrost

boundary of the TP. The Tanggula (TGL) monitoring station is located in the hinterland of the TP and is characterized by a continuous and cold permafrost zone (Yao et al., 2011; Zhao et al., 2021). There are two stations in the Ngoring Lake basin, which is located in the Yellow River source area east of the TP (Li et al., 2017). One grassland station (NLGS) is located on a flat surface, and the other observation station (NLTS) is located on the lakeside beside the lakeshore tower station (Li et al., 2020; Li et al.,

2021). The in situ data of D105 and NPAM come from the Coordinated Enhanced Observing Period Asia-Australia Monsoon Project (CAMP) on the Tibetan Plateau (CAMP/Tibet) (Ma et al., 2009;

Zhong et al., 2010; Ma et al., 2014). Plausible value checks, time consistency checks and internal consistency checks were applied to ensure the accuracy and reliability of the observations. The original sampling data with high frequency were uniformly processed into 30 min and hourly average data by data loggers (e.g., CR3000, CR1000) (Campbell Sci., USA). To retain the observations in their original form as much as possible, no further postprocessing processes are taken, except for replacing outliers with missing values (NaN). Meanwhile, periodic inspection, maintenance and calibration are carried out by professional engineers at all stations.

**Table 2.** Basic information for the in situ observation stations over the Tibetan Plateau.

| Site | Lat(°N) | Lon(°E) | Altitude(m) | Land cover | Instrument | Frequency |
|------|---------|---------|-------------|------------|------------|-----------|
| BJ | 31.37 | 91.90 | 4509 | Alpine meadow | CM21, Kipp & Zonen | 1 h |
| D105 | 33.06 | 91.93 | 5039 | Alpine sparse grassland | CM21, Kipp & Zonen | 1 h |
| NPAM | 31.93 | 91.71 | 4620 | Alpine meadow and grassy marshland | CM21, Kipp & Zonen | 1 h |
| QOMS | 28.36 | 86.95 | 4298 | Gravel and sparse meadow | CNR1, Kipp & Zonen | 1 h |
| SETORS | 29.77 | 94.73 | 3327 | Alpine meadow | CNR1, Kipp & Zonen | 1 h |
| MAWORS | 38.41 | 75.05 | 3668 | Alpine desert | NR01, Kipp & Zonen | 1 h |
| NADORS | 33.39 | 79.70 | 4270 | Alpine desert | NR01, Kipp & Zonen | 1 h |
| NAMORS | 30.77 | 90.98 | 4730 | Alpine steppe | NR01, Vaisala | 1 h |
| NLGS | 34.91 | 97.55 | 4280 | Flat Grassland | CNR4, Kipp & Zonen | 0.5 h |
| NLTS | 34.91 | 97.57 | 4275 | Water | CNR4, Kipp & Zonen | 0.5 h |
| XDT | 35.72 | 94.13 | 4538 | Alpine meadow | CM3, Kipp & Zonen | 0.5 h |
| TGL | 33.07 | 91.94 | 5100 | Alpine sparse meadow | CM3, Kipp & Zonen | 0.5 h |


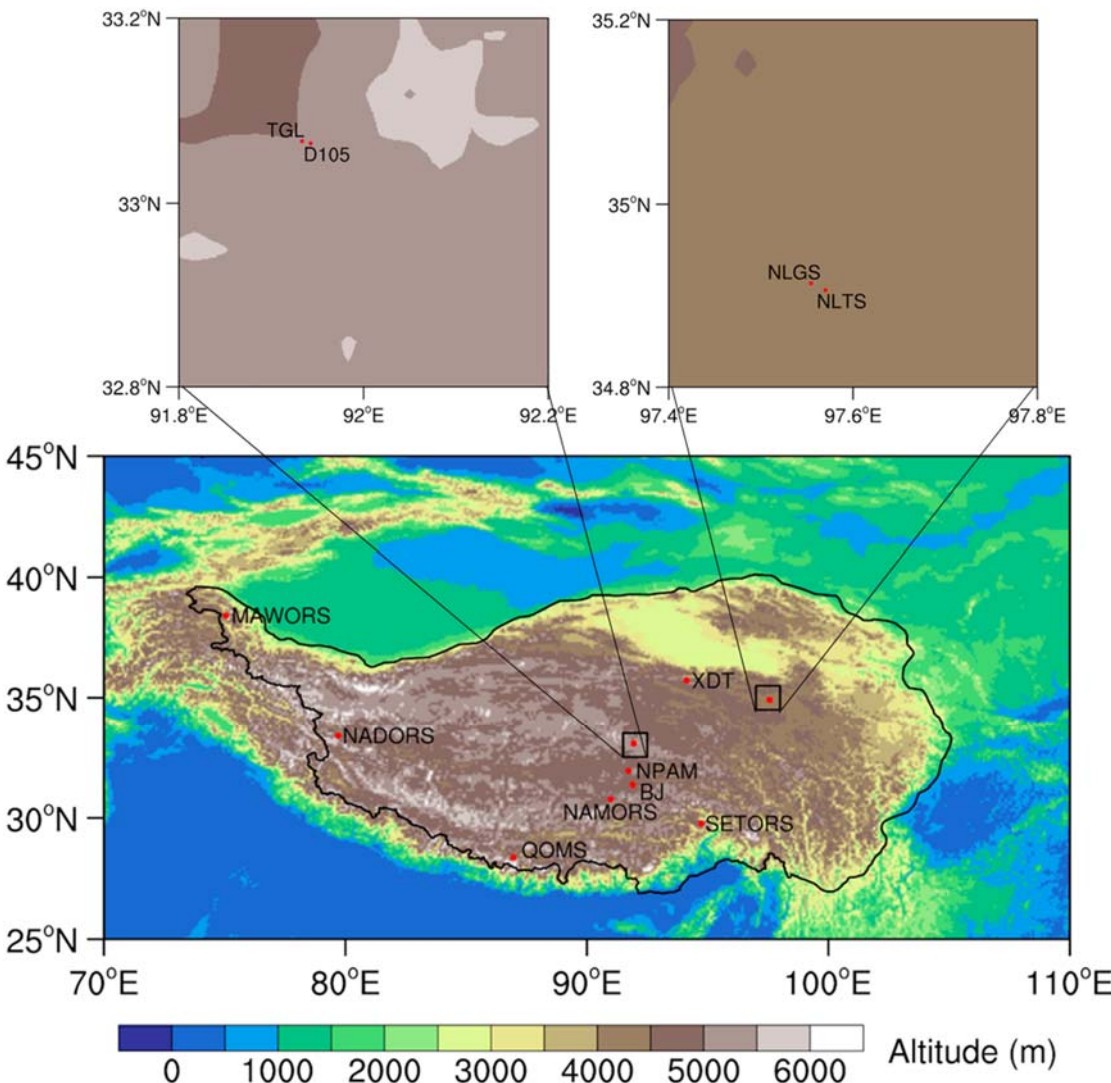

**Figure 1.** Locations of the twelve in situ observation stations over the TP. The legend of the color map indicates the elevation above mean sea level in meters.

**3 Methodology**

The effects caused by ozone, aerosol, water vapor, Rayleigh scattering, permanent gases, clouds and terrain are comprehensively taken into account in the improved parameterization scheme. More importantly, the DSR varies with altitude, surface slope and aspect, and the multiple actions of cloud and topography factors on DSR have been neglected in many previous studies. The all-sky DSR estimation method is divided into two groups, one for clear-sky conditions and the other for cloudy-sky

conditions. The main steps of the method and related key variables are shown in Fig. 2.

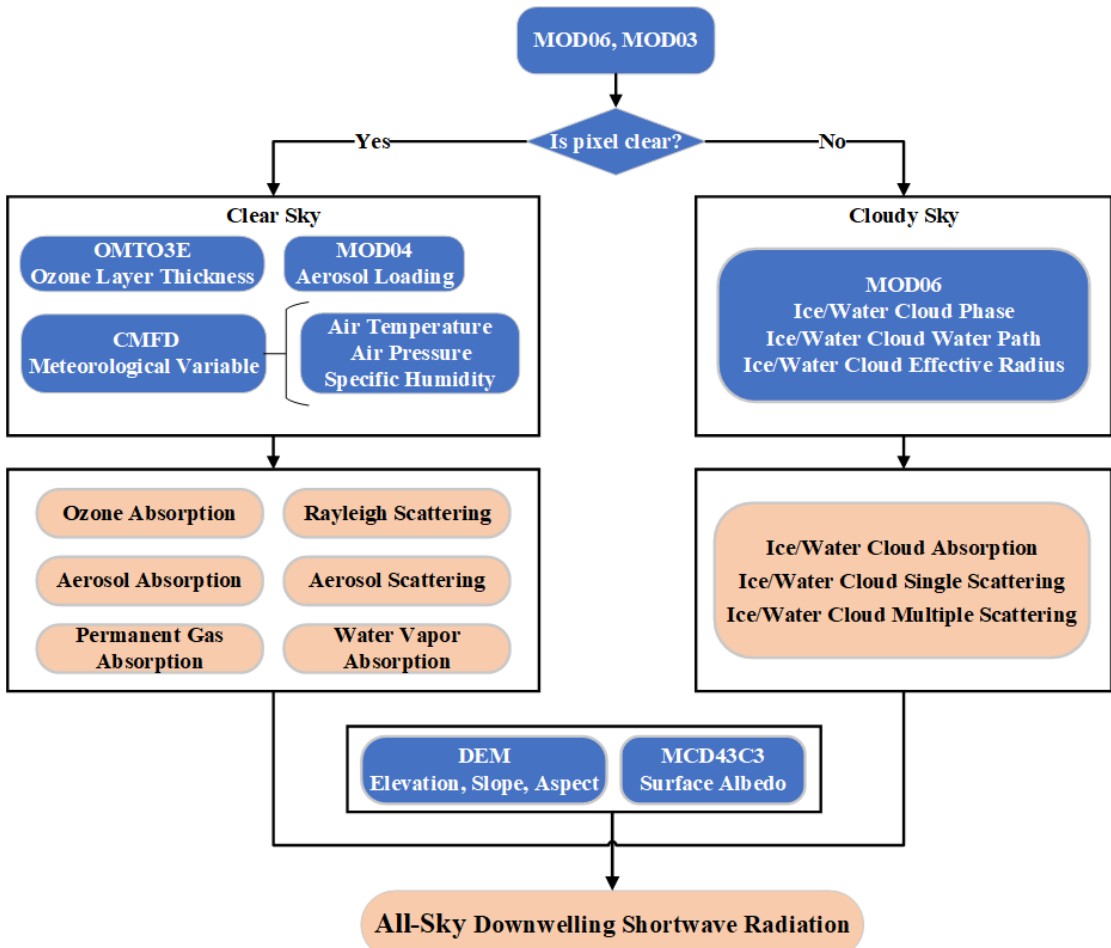

**Figure 2.** Flowchart for estimating all-sky DSR.

### 3.1 Clear-Sky Scheme

The DSR under clear-sky conditions ($DSR_{clr}$) can be calculated as the sum of three components: direct
(beam) radiation ($S_{b,clr}$), diffuse radiation ($S_{d,clr}$), and reflected insolation from the surrounding terrain
($S_{r,clr}$).

$$DSR_{clr} = S_{b,clr} + S_{d,clr} + S_{r,clr} = S_0(\tau_{b,clr} + \tau_{d,clr} + \tau_{r,clr}), \tag{1}$$

where $S_0$ denotes the horizontal extraterrestrial solar insolation, which may slightly change with the
earth-sun distance throughout the year. In addition, $\tau_{b,clr}$ is the direct radiative transmittance; $\tau_{d,clr}$ is
the diffuse radiative transmittance; and $\tau_{r,clr}$ is the reflectance radiative transmittance.

### 3.2 Cloud-Sky Scheme

DSR under cloudy-sky conditions ($DSR_{cld}$) can be divided into four items as follows:

$$DSR_{cld} = S_{b,cld} + S_{d,cld} + S_{am,cld} + S_{r,cld}$$

$$= S_0\tau_{b,cld} + S_0\tau_{d,cld} + S_0\big(\tau_{b,cld} + \tau_{d,cld}\big)\frac{\rho_{a,cld}\rho_g}{1-\rho_{a,cld}\rho_g} + S_0\tau_{r,cld}, \tag{2}$$

where the first, second and fourth items are the direct solar irradiance ($S_{b,cld}$), diffuse solar irradiance ($S_{d,cld}$), and reflected solar irradiance ($S_{r,cld}$) under cloudy conditions, respectively. The third item is the ambient solar irradiance caused by the interactions between the surface and atmosphere ($S_{am,cld}$). $\tau_{b,cld}$ is the direct radiative transmittance; $\tau_{d,cld}$ is the diffuse radiative transmittance; $\tau_{r,cld}$ is the reflectance radiative transmittance; and $\rho_{a,cld}$ is the atmosphere hemispherical albedo under

cloudy-sky conditions. $\rho_g$ is the ground surface albedo.

The variations in elevation, slope and aspect of the land surface are considered for the above radiative transmittance. A detailed description of $\tau_{b,clr}$, $\tau_{d,clr}$, $\tau_{r,clr}$, $\tau_{b,cld}$, $\tau_{d,cld}$, $\tau_{r,cld}$, $\rho_{a,cld}$ and $\rho_g$ is presented in Appendix A.

## 4 Results and Discussions

Considering the integrity and temporal continuity of the available data, the data of the BJ, D105, NPAM and SETORS stations in 2008, the data of the QOMS station in 2008 and 2015, and the data of the remaining seven stations in 2015 are used for validation. To ensure the reliability of the validation, first, the outliers in the ground-based observations were removed by considering the valid range and time continuity. Then, the hourly data were smoothed to 30 minutes to match the satellite overpass time

and the station observation time (Huang et al., 2016b). The root-mean-square error (RMSE), mean bias (MB), mean absolute error (MAE) and Pearson correlation coefficient (R) are used to evaluate the performance of the radiation parameterization scheme.

### 4.1 Validation against in situ measurements

As shown in Fig. 3a and 3b, at the instantaneous scale, the RMSE and R of the 1 km DSR under

clear sky are 105.34 W m$^{-2}$ and 0.76, respectively, while those of the 1 km all-sky DSR are 158.19 W m$^{-2}$ and 0.70, respectively. The validation results of this study are not as good as those in other plain areas, where RMSE and R are usually approximately 60 W m$^{-2}$ and 0.9 under clear skies, while those of all-sky conditions are approximately 100 W m$^{-2}$ and 0.9, respectively. Nevertheless, considering the unique climate characteristics of the TP and compared with the existing DSR products and algorithms

(see Section 4.2 and Section 4.4 for details), the accuracy of the results is within an acceptable range.

Roupioz et al. (2016) estimated all-sky solar radiation at an instantaneous timescale based on MODIS products, but the retrievals were validated using only BJ, QOMS and NAMORS stations. In their study, the RMSE, MB and R of BJ station were 225.5 W m$^{-2}$, 120.1 W m$^{-2}$ and 0.51, respectively; the RMSE, MB and R of QOMS station were 117.1 W m$^{-2}$, 13.0 W m$^{-2}$ and 0.74, respectively; and the RMSE, MB and R of NAMORS station were 203.5 W m$^{-2}$, 39.5 W m$^{-2}$ and 0.55, respectively. Table 3 shows that the accuracy of our DSR estimation is better than Roupioz's retrievals.

Representativeness errors of point-scale measurements can affect the validation results of instantaneous DSR estimations to some extent. The insufficient spatial representation of point-scale observations can be partly compensated by lowering their temporal resolution (Hakuba et al., 2013; Huang et al., 2016b). Therefore, the DSR estimation results were also validated at ten-day and monthly timescales. It is upscaled to ten-day and monthly timescales via averaging by instantaneous values. There are three 10-day periods within 1 month, which can be defined as follows: from the first to the 10th, from the 11th to the 20th, and from the 21st to the end of every month. Obviously, the estimation of DSR at a longer timescale shows more reasonable agreement with the in situ measurements compared with the instantaneous DSR estimations (Fig. 3c and Fig. 3d).

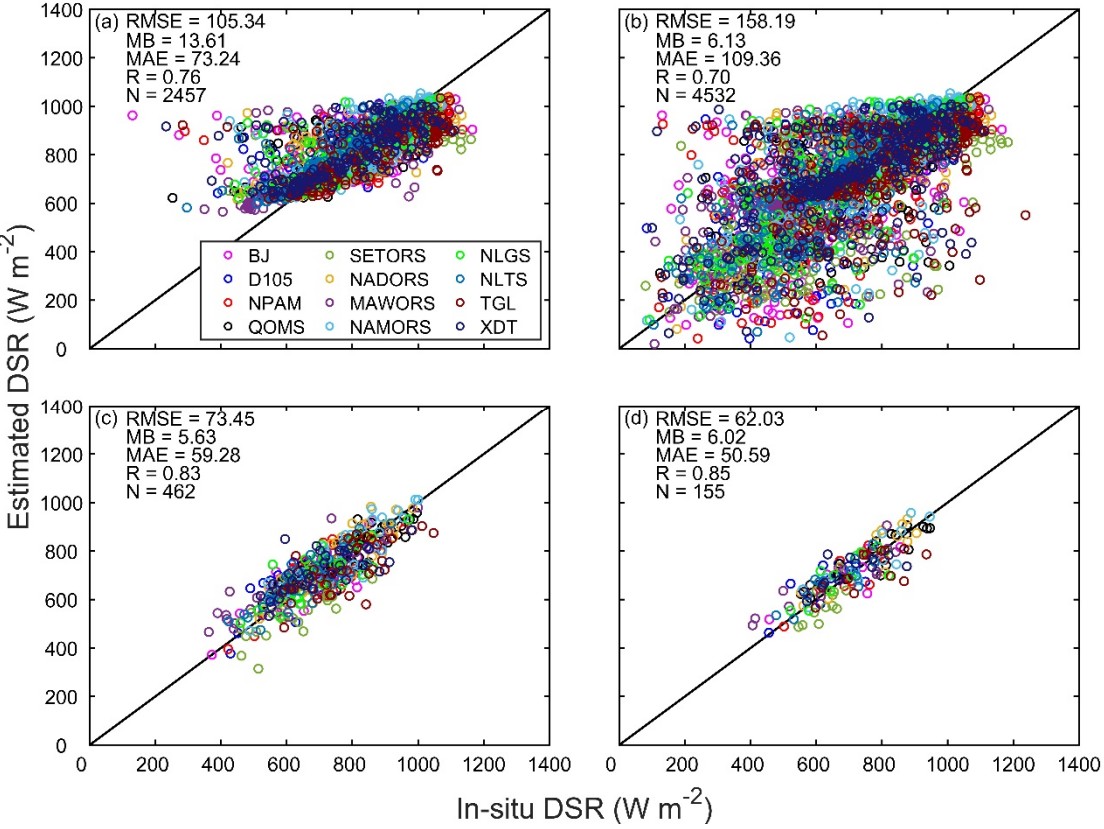

**Figure 3.** Validation results for the estimated DSR at (a and b) instantaneous scale, (c) ten-day scale and (d)

monthly scale. Scatter plots (a) and (b) show the validation results of instantaneous DSR under clear sky and all-sky conditions, respectively. N indicates the number of points. The legend with different colors denotes the twelve stations involved in the validation. The units of RMSE, MB and MAE are W m$^{-2}$.

The corresponding statistical indices for the twelve stations in this study are listed in Table 3. Since there is usually a distinctness between DSR estimation under clear-sky and cloudy-sky conditions, the statistics of specific stations are always related to the overall cloud fraction. Therefore, the proportion of cloud cover days (CCD) at each station is also listed in the table. Zhong et al. (2019b) estimated all-sky solar radiation on a ten-day timescale based on MODIS products over the TP, while their method needed to obtain ground-based measurements in advance for local calibration. We find that compared to the statistics presented at the D105, QOMS, and SETORS stations, the accuracies of our method are on average slightly higher.

As illustrated in Table 3, the best validation results occurred at the QOMS station, showing the lowest RMSE, the MB of a smaller absolute value, and the higher R, due to the extremely low CCD over there (~19.83%), whereas the poorer performance occurred at the SETORS and TGL stations, according to the validation results on various time scales. The SETORS station is located in the valley near the southeastern TP, surrounded by dense vegetation (mainly evergreen needle-leaved trees and alpine meadows) and is close to the southern water vapor transport channel. Accordingly, many precipitation events occur here, with a maximum CCD (~72.85%) among the twelve sites. The TGL station lies on the north side of the Tanggula Mountains, surrounded by numerous glaciers and deep snow cover, which can persist for many days (Xu et al., 2017; Zhou et al., 2018). Because the snow/ice cover beneath the clouds is difficult to identify from satellite signals, there is great uncertainty in the corresponding retrievals of cloud microphysical parameters, which may lead to low accuracy of the estimation results. In addition, previous studies have shown that snow cover will result in the underestimation of DSR (Pinker et al., 2007; Huang et al., 2016a), which is also indicated by the large negative MB of the TGL site compared with other stations.

**Table 3.** Summary statistics of the validation results for each station on different timescales.

| Site | Instantaneous timescale | | | | Ten-day timescale | | | | Monthly timescale | | | | CCD |
| | RMSE (W m$^{-2}$) | MB (W m$^{-2}$) | R | N | RMSE (W m$^{-2}$) | MB (W m$^{-2}$) | R | N | RMSE (W m$^{-2}$) | MB (W m$^{-2}$) | R | N | |
| --- | --- | --- | --- | --- | --- | --- | --- | --- | --- | --- | --- | --- | --- |
| BJ | 179.44 | 11.41 | 0.66 | 359 | 66.20 | 13.91 | 0.84 | 36 | 56.01 | 14.54 | 0.81 | 12 | 49.58% |
| D105 | 162.87 | 32.47 | 0.67 | 359 | 76.69 | 33.23 | 0.73 | 36 | 67.43 | 33.80 | 0.73 | 12 | 54.02% |

| | | | | | | | | | | | | |
|---|---|---|---|---|---|---|---|---|---|---|---|---|
| NPAM | 177.57 | -3.90 | 0.63 | 358 | 67.63 | -4.28 | 0.82 | 36 | 51.90 | -3.75 | 0.82 | 12 | 53.46% |
| QOMS | 112.33 | 5.04 | 0.74 | 689 | 56.49 | 6.38 | 0.90 | 69 | 49.76 | 6.41 | 0.91 | 23 | 19.83% |
| SETORS | 183.33 | -49.51 | 0.67 | 302 | 94.17 | -49.48 | 0.67 | 33 | 64.89 | -44.04 | 0.74 | 12 | 72.85% |
| MAWORS | 167.41 | 28.51 | 0.71 | 350 | 83.27 | 27.08 | 0.90 | 36 | 72.94 | 27.32 | 0.92 | 12 | 55.62% |
| NADORS | 129.88 | 19.48 | 0.78 | 318 | 66.20 | 17.59 | 0.89 | 36 | 58.30 | 18.20 | 0.90 | 12 | 35.07% |
| NAMORS | 150.62 | 18.30 | 0.72 | 342 | 65.60 | 13.66 | 0.88 | 36 | 55.92 | 13.42 | 0.89 | 12 | 40.27% |
| NLGS | 141.53 | 11.26 | 0.77 | 365 | 66.51 | 10.81 | 0.81 | 36 | 56.48 | 11.02 | 0.80 | 12 | 46.58% |
| NLTS | 136.29 | 24.63 | 0.79 | 360 | 62.80 | 22.01 | 0.86 | 36 | 51.55 | 23.81 | 0.87 | 12 | 59.45% |
| XDT | 183.08 | 17.84 | 0.63 | 365 | 81.41 | 17.95 | 0.72 | 36 | 70.48 | 18.02 | 0.70 | 12 | 51.23% |
| TGL | 188.98 | -46.64 | 0.58 | 365 | 97.70 | -46.52 | 0.72 | 36 | 87.80 | -46.92 | 0.66 | 12 | 45.63% |

## 4.2 Comparison among different DSR products

To further evaluate the reliability of our DSR estimates, several existing widely used DSR products were selected for comparison based on the same in situ observations used in Section 4.1. Among these products, there are remotely sensed and reanalysis DSR products, namely, Clouds and the Earth's Radiant Energy System Synoptic (CERES_SYN) surface fluxes (Loeb et al., 2013), Global Energy and Water Exchanges Surface Radiation Budget (GEWEX_SRB) datasets (Zhang et al., 2014), MODIS DSR product (MCD18A1) (Wang et al., 2020) and the fifth generation reanalysis (ERA5) from the European Centre for Medium-Range Weather Forecasts (ECMWF) (Hans et al., 2019). In addition, Letu et al. (2022) produced a high-resolution (5 km, 10 min) DSR dataset (short for "H-8_EAP" in our study) under all-sky conditions from 2016 to 2020 in the East Asia–Pacific region based on the next-generation geostationary satellite Himawari-8/AHI, which was also selected for comparison. At present, the latest in situ data in this study are in 2016, and the Himawari-8 satellite cannot observe the western part of the TP. Therefore, six stations (BJ, QOMS, SETORS, NAMORS, NLGS and NLTS) in 2016 are selected for comparison with the H-8_EAP DSR dataset.

The spatial resolutions of MCD18A1 and ERA5 are 1 km and 25 km, respectively. CERES_SYN and GEWEX_SRB have a spatial resolution of 100 km. It is known that spatial mismatch may incur errors in the validation results, so our results at the original scale of 1 km were aggregated to the corresponding spatial resolution of the above products. The temporal resolution of MCD18A1 is instantaneous. GEWEX_SRB has a temporal resolution of 3 hours and ERA5 has a temporal resolution of 1 hour. CERES_SYN products have two temporal resolutions of 1 hour and 3 hours. The abovementioned DSR products and the estimated DSR of this study were temporally matched to 10:30

local time for mutual comparison (Zhong et al., 2019b).

**Table 4.** Comparison with existing DSR products on different timescales in terms of accuracy.

| Product name | Instantaneous timescale | | | Ten-day timescale | | | Monthly timescale | | | Spatial resolution |
|---|---|---|---|---|---|---|---|---|---|---|
| | RMSE (W m$^{-2}$) | MB (W m$^{-2}$) | R | RMSE (W m$^{-2}$) | MB (W m$^{-2}$) | R | RMSE (W m$^{-2}$) | MB (W m$^{-2}$) | R | |
| MCD18A1 | 233.47 | -76.43 | 0.60 | 147.04 | -74.60 | 0.72 | 130.24 | -74.17 | 0.74 | 1 km |
| This study | 152.13 | 5.23 | 0.72 | 77.24 | 7.35 | 0.82 | 63.79 | 7.25 | 0.84 | |
| H-8_EAP | 197.89 | -52.47 | 0.66 | 140.67 | -57.07 | 0.67 | 125.70 | -62.74 | 0.73 | 5 km |
| This study | 140.54 | 23.64 | 0.77 | 82.67 | 21.54 | 0.78 | 71.48 | 14.97 | 0.81 | |
| ERA5 | 165.67 | -20.59 | 0.65 | 88.06 | -21.44 | 0.82 | 74.19 | -21.06 | 0.86 | 25 km |
| This study | 135.11 | 15.67 | 0.77 | 75.01 | 15.24 | 0.83 | 67.12 | 15.75 | 0.83 | |
| CERES_SYN_1h | 146.64 | -46.70 | 0.75 | 84.27 | -47.93 | 0.86 | 73.25 | -47.53 | 0.89 | |
| CERES_SYN_3h | 160.50 | -78.30 | 0.74 | 107.13 | -79.48 | 0.85 | 98.67 | -79.06 | 0.88 | 100 km |
| GEWEX_SRB | 194.45 | -118.56 | 0.68 | 143.68 | -119.71 | 0.80 | 135.54 | -119.21 | 0.83 | |
| This study | 132.84 | 2.79 | 0.77 | 70.84 | 2.18 | 0.84 | 61.33 | 2.70 | 0.85 | |

As summarized in Table 4, the RMSE range of these DSR products is approximately 150~230 W m$^{-2}$ at the instantaneous scale. At the ten-day scale, the RMSE range is approximately 80~150 W m$^{-2}$. At the monthly scale, the RMSE range is approximately 70~130 W m$^{-2}$. The MB range of these DSR products is -120 ~ -20 W m$^{-2}$ at three temporal scales. These large spans of RMSE and MB indicate that the current DSR products still have great uncertainties over the TP. The RMSE ranges of this study at three temporal scales are 132~152, 70~82 and 61~71 W m$^{-2}$. The MB range of this study is 3 ~ 24 W m$^{-2}$ at three temporal scales. The estimates of this study show a smaller RMSE, lower absolute value MB and comparable R values at the corresponding spatial and temporal scales. This means that the derived DSR based on the proposed method performs better than other DSR products over the TP.

In addition, it is noted that the accuracies of all datasets have been appreciably improved with increasing timescale. This is because the 3D radiative transfer effects and complexity of clouds can be significantly reduced and the spatial representativeness of ground-based measurements can be significantly enhanced through temporal averaging (Huang et al., 2016b; Huang et al., 2016a). A phenomenon in which the RMSE of this study has been improved with incremental space scales at three time scales is also found, while the variations are relatively small at the ten-day and monthly scales. This may be because the time mismatch between satellite observations and surface measurements can be partly decreased by inherent averaging in the upscaling of spatial resolution

(Tang et al., 2019).

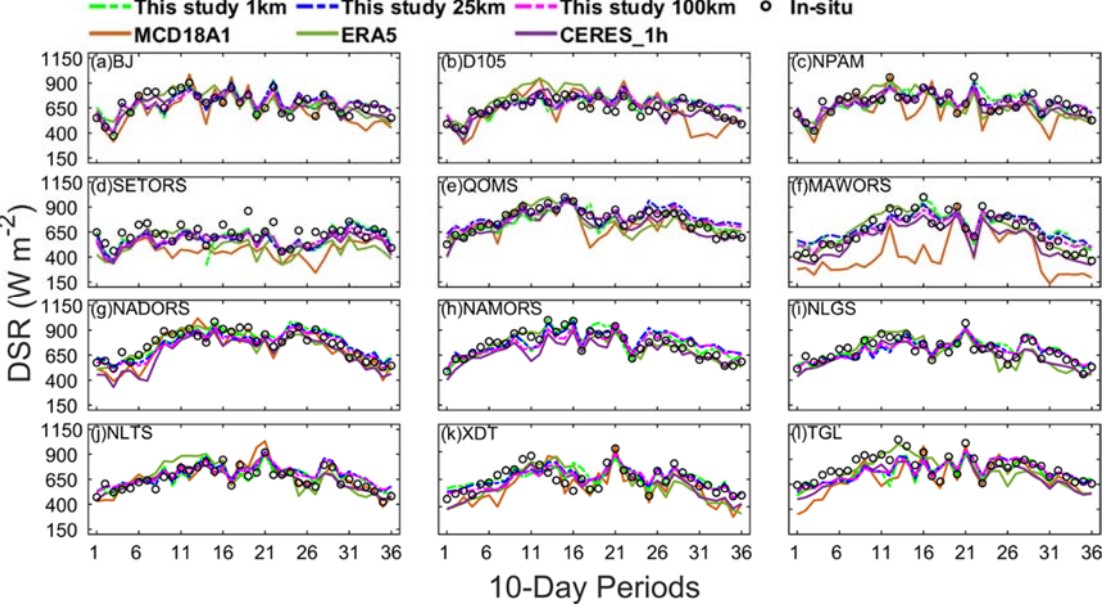

**Figure 4.** Intercomparison of time series of DSR among MCD18A1, ERA5, CERES_SYN_1 h, and this study at (a) BJ, (b) D105, (c) NPAM, (d) SETORS, (e) QOMS, (f) MAWORS, (g) NADORS, (h) NAMORS, (i) NLGS, (j) NLTS, (k) XDT, and (l) TGL stations on a ten-day timescale. The circle denotes in situ data.

DSR products with relatively high accuracy, which correspond to three spatial resolutions of 1 km, 25 km and 100 km, are selected for comparison with the estimated DSR in this study in terms of temporal variation characteristics (Fig. 4). The time series of MCD18A1 at NAMORS and NLGS stations are not displayed because there are many missing values in MCD18A1 at these two stations. It can be seen that six selected DSR showed a quasi-convex shape in one year at all stations except SETORS. There are some fluctuations in DSR during the summer monsoon period due to the high frequency of clouds and precipitation. Almost all six selected DSRs showed relatively smooth variation at SETORS compared with other stations, which demonstrated a large variation with time. The dynamic range (defined as the difference between the maximum and the minimum in a year) of MCD18A1 is the largest, while ERA5, CERES_SYN_1 h and this study show similar dynamic ranges. Compared with other products, the derived DSR of this study is more consistent with the in situ observations at each station, and all show similar temporal change trends.

It should be noted that the six selected DSRs are not consistent with the in situ observations at the SETORS station, especially in the monsoon period in which obvious underestimation can be found. Cloud and precipitation occurrence frequencies generally reach peaks during the monsoon period over the TP. Compared with other regions of the TP, not only higher cloud amounts and frequencies but also

higher precipitation intensities and frequencies are found in the southeastern TP, where the SETORS station is located (Zhao et al., 2019a; Kukulies et al., 2020). Convective clouds appear most frequently over the TP near noon, and thus, the DSR may also partially come from the high diffuse radiation caused by cloud scattering in addition to direct radiation (Fujinami et al., 2005; Li et al., 2008; Yang et al., 2010a). It is still difficult to reflect the 3D radiation effect of clouds, although this study has considered the scattering of clouds and thus may lead to underestimation of DSR. The microphysical processes of convective clouds generally include mixed-phase processes inside clouds (Fu et al., 2020). Nevertheless, only a single phase can be diagnosed by satellite-based cloud products, which may significantly influence the retrieval accuracy of DSR (Platnick et al., 2003; Platnick et al., 2017). In addition, the SETORS station is flat with grass cover, while its surroundings are valley and dense evergreen needle-leaved trees. The domes of instruments are vulnerable to the contamination of precipitation, and further influence the spatial representativeness of in situ stations. Hence, some errors may be introduced due to the inadequate spatial representativeness of point-scale measurements compared with the coarse resolution of satellite images.

## 4.3 Spatiotemporal variations in surface downward shortwave radiation

Based on the above analysis, CERES_SYN_1 h and ERA5 performed better than the other DSR products. To better investigate the spatiotemporal variations in the estimated DSR over the TP, the seasonal spatial distribution of DSR generated from CERES_SYN_1 h, ERA5 and this study in 2008 are collected and compared in Fig. 5. In general, the three mentioned DSR provide similar seasonal radiation patterns, i.e., the DSR values are higher in spring and summer and lower in autumn and winter. This phenomenon can also be found in the monthly mean DSR variation over the TP (Fig. 5m). The DSR increased from a minimum value in January to a maximum value in April. The formation of this pattern is primarily controlled by the north-south movement of the subsolar point.

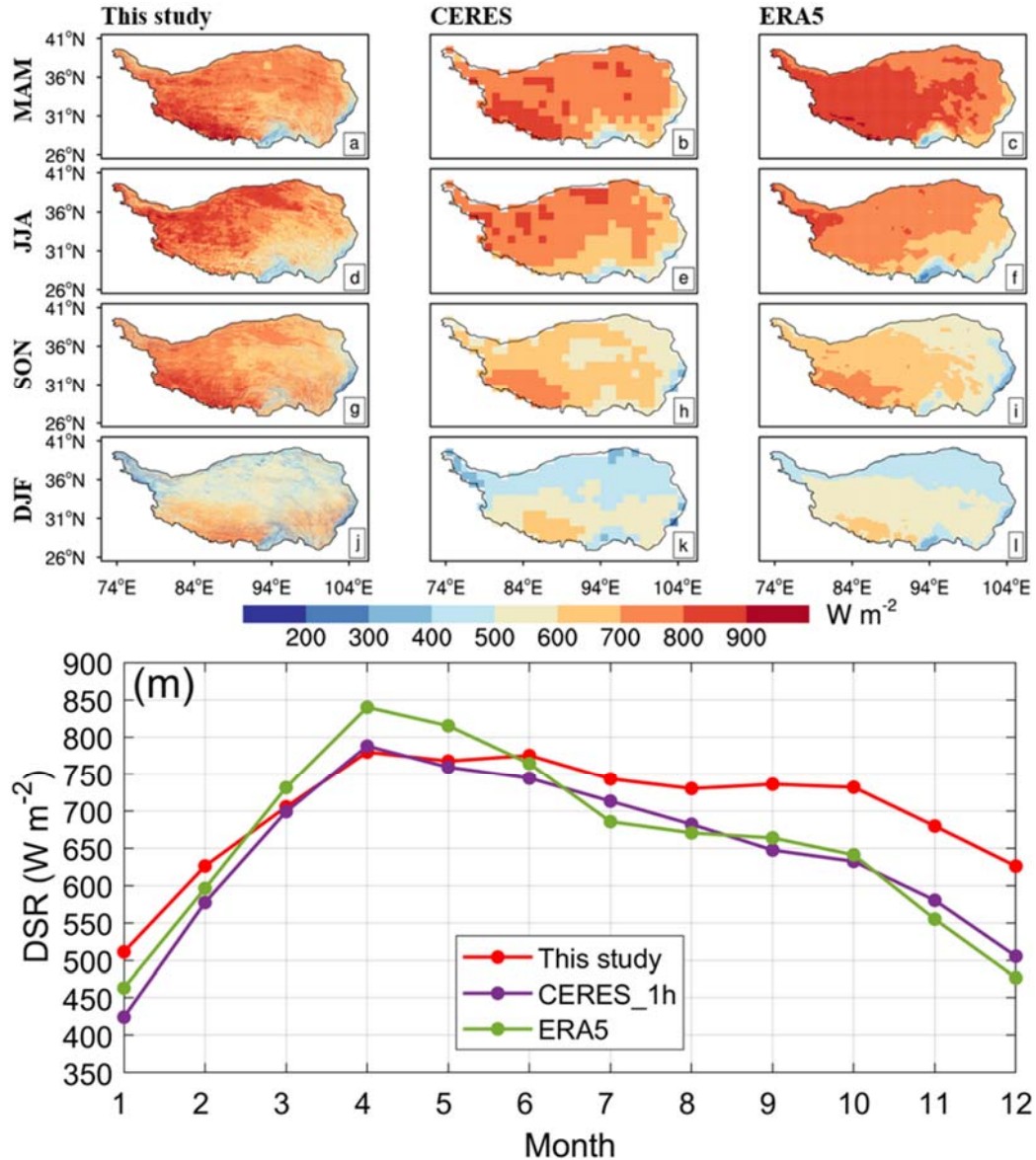

**Figure 5.** Spatial distribution of DSR from this study (left), CERES_SYN_1 h (center), and ERA5 (right) for four seasons in 2008 over the TP. The first to fourth rows represent spring (March, April and May), summer (June, July and August), autumn (September, October and November), and winter (December, January and February), respectively. The bottom panel (m) shows the comparison of monthly temporal variation of spatial mean DSR estimates from this study, CERES_SYN_1 h, and ERA5 over the TP.

It should be pointed out that the radiation texture of this study is rather clear due to the higher resolution (1 km), and more details of DSR variations can be captured. The high values of DSR are mostly located in the western TP. This can be explained by the fact that the western TP, with arid and semiarid climate characteristics, has a higher altitude than the eastern TP, and thus, less radiation attenuation occurred. At the same time, the southern margin of the TP and the eastern margin of the TP near the Sichuan Basin are always low-value areas of DSR. The south edge of the TP is a water vapor

transport channel associated with the South Asian monsoon, and the frequencies of clouds and rainfall are high. The eastern edge of the TP near the Sichuan Basin has a very low altitude (~ 1800 m) and is often covered by stratiform clouds. Accordingly, strong solar radiation attenuation occurred in these two regions.

The difference among the three mentioned DSRs is also displayed in Fig. 5. The high value of DSR appears in the southwestern TP in spring, but the high value of ERA5 covers a large area and even extends to the Tanggula Mountains (Fig. 5a-c). The overall DSR pattern over the TP shows a decreasing trend from northwest to southeast in summer, but the high value in the Qaidam Basin is not found in ERA5 (Fig. 5d-f). In autumn, the high value of DSR is concentrated in the southwestern TP, showing a spatial pattern of high-west and low-east (Fig. 5g-i). In winter, the DSR reaches the minimum of the year and shows a spatial distribution of high-south and low-north over the TP (Fig. 5j-l). However, the DSR derived from this study is generally higher than that of the other two products. The monthly temporal variation in the spatial mean DSR over the TP also indicates a similar phenomenon (Fig. 5m). The spatial mean DSR of ERA5 is higher than those of the other two DSR products in spring, and the spatial mean DSR estimated in this study is higher than those of the other two radiation products in autumn and winter.

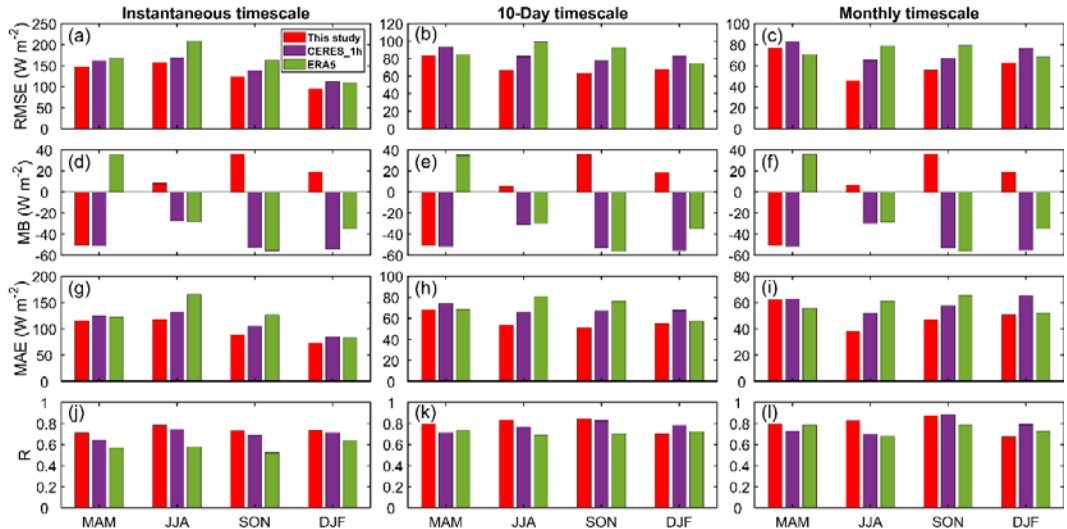

**Figure 6.** RMSE (a-c), MB (d-f), MAE (g-i) and R (j-l) between in situ observations and DSR estimates from this study (red bar), CERES_SYN_1 h (purple bar), and ERA5 (green bar) products in four seasons.

To further understand the difference between the three mentioned DSRs, the corresponding statistical indices for the four seasons are presented in Fig. 6. At all temporal scales in spring, ERA5 shows a positive bias, while the other two DSRs show a negative bias. In summer, autumn and winter,

the DSR estimated from this study shows positive bias, while the other two products show negative

bias. This explains the above phenomenon. However, this study is significantly lower than the other

two DSR products in terms of the absolute value of MB. Particularly, it can be clearly seen that in all

seasons and temporal scales, not only in MB but also in terms of RMSE and MAE, this study shows the

lowest values, and in terms of R, this study is comparable to or higher than the CERES_SYN_1 h and

ERA5 products. Similar comparison and verification results can also be found in Table 4. In addition,

the spatial distribution of this study is similar to that in a previous study by Zhong et al. (2019b).

Therefore, it is not difficult to conclude from the above analysis that the DSR patterns of this study are

reasonable enough, at least showing advantages over other products in terms of spatial resolution with

relevant details.

**4.4 Evaluation of cloud multiscattering and topographic effects**

To evaluate the effects of cloud multiscattering and complex topography, the accuracy of the DSR

derived with and without considering terrain factors and cloud multiple scattering on different temporal

scales were compared (Table 5). Here, four simple cases were designed. Both terrain factor and cloud

multiple scattering are not included in Case 1; Case 2 only considers terrain factor, and Case 3 only

considers cloud multiple scattering. Case 4 is the method adopted in this study; that is, both terrain

factor and cloud multiple scattering are taken into account.

**Table 5.** Comparison between DSR estimation with and without considering cloud multiple scattering and terrain factors on different timescales in terms of accuracy.

|  |  | Case1 | Case2 | Case3 | Case4 |
|---|---|---|---|---|---|
| Instantaneous timescale | RMSE (W m$^{-2}$) | 192.90 | 177.77 | 174.52 | 158.19 |
|  | MB (W m$^{-2}$) | 57.23 | 12.04 | 51.58 | 6.13 |
|  | MAE (W m$^{-2}$) | 132.48 | 119.71 | 121.74 | 109.36 |
|  | R | 0.69 | 0.65 | 0.73 | 0.70 |
| Ten-day timescale | RMSE (W m$^{-2}$) | 96.54 | 80.79 | 87.53 | 73.45 |
|  | MB (W m$^{-2}$) | 56.79 | 11.39 | 51.17 | 5.63 |
|  | MAE (W m$^{-2}$) | 77.52 | 63.42 | 70.98 | 59.28 |
|  | R | 0.87 | 0.81 | 0.89 | 0.83 |
| Monthly timescale | RMSE (W m$^{-2}$) | 84.50 | 66.45 | 77.44 | 62.03 |
|  | MB (W m$^{-2}$) | 57.58 | 11.99 | 51.61 | 6.02 |
|  | MAE (W m$^{-2}$) | 69.78 | 53.04 | 63.80 | 50.59 |
|  | R | 0.90 | 0.83 | 0.91 | 0.85 |

As shown in Table 5, the RMSE of case 1 reaches nearly 200 W m$^{-2}$ at the instantaneous scale, nearly 100 W m$^{-2}$ at the ten-day scale, and more than 80 W m$^{-2}$ at the monthly scale, all of which are the highest among the four cases. As mentioned earlier, the estimated DSR of the SETORS station is more vulnerable to clouds. The RMSE of the SETORS station is reduced by 15%-19% when cloud multiple scattering is considered. Hence, ignoring the multiple scattering of clouds may lead to large errors in the case of high cloud cover. The verification results are improved when multiple cloud scattering and varying topography are introduced, and the RMSE is reduced by 8%-25%. Obviously, Case 4 shows the lowest RMSE, MB, MAE, and comparable R values compared with the other three cases. This reflects that when estimating DSR under all-sky conditions over the TP, the effects of terrain and cloud multiscattering cannot be simply ignored.

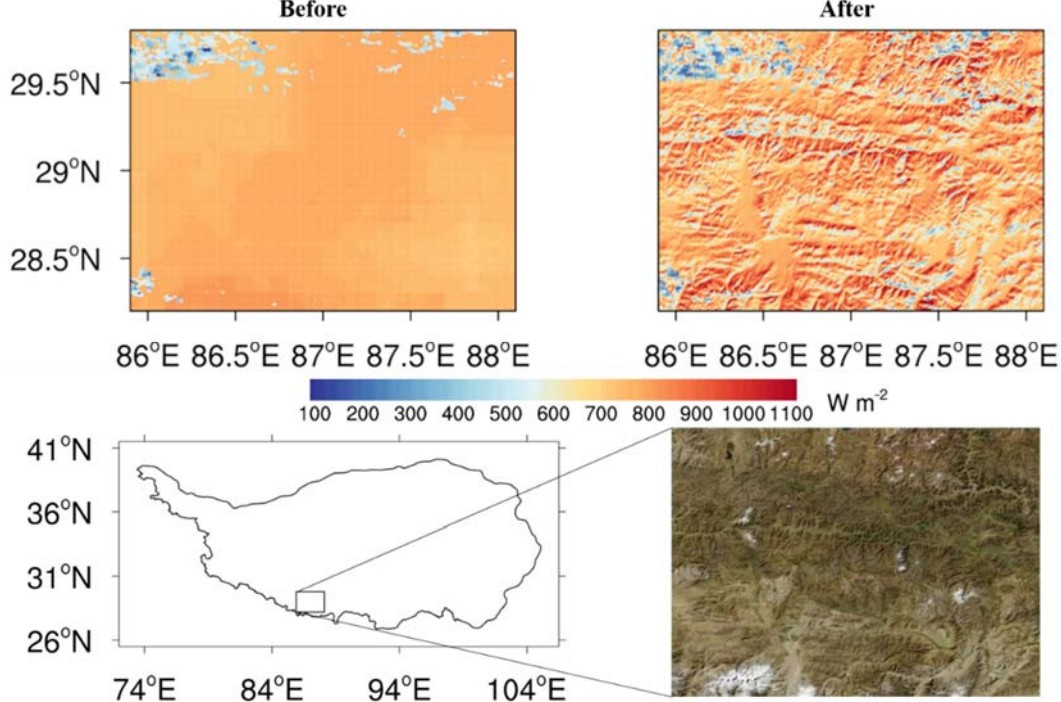

**Figure 7.** DSR estimated before terrain correction (left) and after terrain correction (right) over the TP at 10:30 LT on 10 January 2008.

To show the impact of varying topography on DSR, the spatial DSR pattern in a subarea of the TP before and after terrain correction is shown under relatively clear-sky conditions (Fig. 7). Before terrain correction, the value of DSR is uniform, and the spatial texture is relatively smooth. The majority of the selected areas show relatively fixed values (~ 750 W m$^{-2}$), except for the parts covered by clouds, which show obviously low values. In contrast, the DSR values show high spatial dynamics, and it is

easy to identify some subtle changes. The spatial gradient of DSR on the sunny and shady slope hillsides is obvious, and the higher parts receive more solar radiation. This is consistent with the surface features shown by the satellite images in the lower right corner.

**4.5 Sensitivity analysis**

The accuracy of the parameterization scheme depends on the quality of the input data to some extent. To further understand the effect of uncertainties in input variables on the accuracy of the DSR retrieval scheme, sensitivity analysis of the DSR to input variables is conducted (Fig. 9 and Fig. 10). As shown in Fig. 8, three points located in the west, north central, and southeast of the TP are randomly selected for sensitivity tests. The average of each input variable (including air temperature Tair, air pressure Pair, specific humidity SH, ozone layer thickness, aerosol optical depth AOD, surface albedo, cloud effective radius CER and cloud water path CWP) for three randomly selected points is selected as the default value.

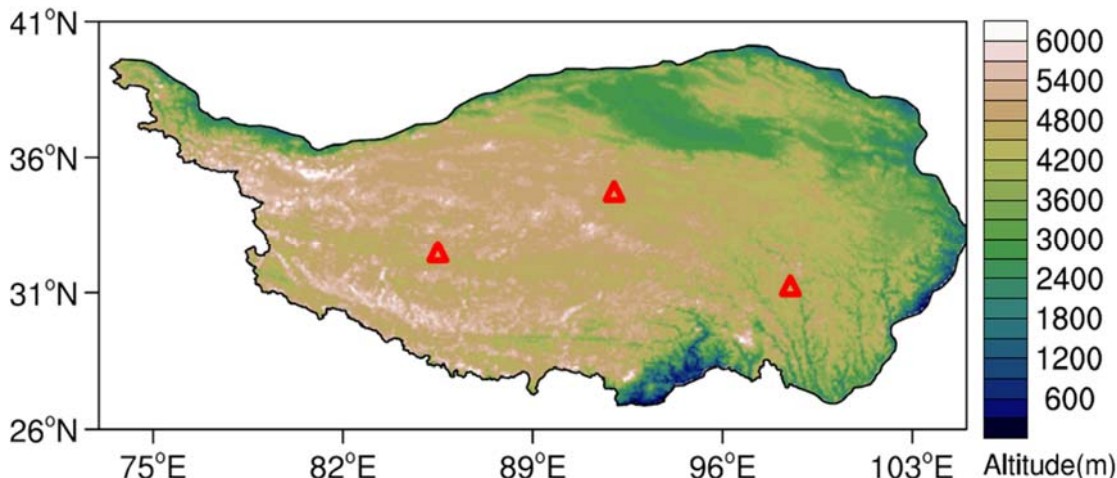

**Figure 8.** Locations of the three points (marked by red triangles) used to carry out sensitivity tests of the input data. The legend of the color map indicates the elevation above mean sea level in meters.

As shown in Fig. 9 and Fig. 10, in terms of changing trend and range, DSR has different responses to fluctuations of each input variable under different sky conditions. The sensitivity test results show that the DSR exhibits a positive correlation with Pair and ozone layer thickness and a negative correlation with Tair under both clear and cloudy conditions, with a nearly linear relationship (Fig. 9a, b, d and Fig. 10a, b, d). The DSR exhibits a negative correlation with SH and AOD with a nonlinear relationship under both clear and cloudy conditions (Fig. 9c,e and Fig. 10c,e). In addition, the DSR exhibits a positive correlation with CER and a nonlinear negative correlation with CWP under cloudy

sky conditions (Fig. 10g and h). However, the DSR exhibits a linear positive correlation with surface
albedo under clear sky conditions, while it displays a nonlinear positive correlation under cloudy sky
conditions (Fig. 9f and Fig. 10f). This phenomenon indicates that multiple scattering effects occur
between the atmospheric medium (aerosols and clouds) and the land surface (Ma et al., 2020).

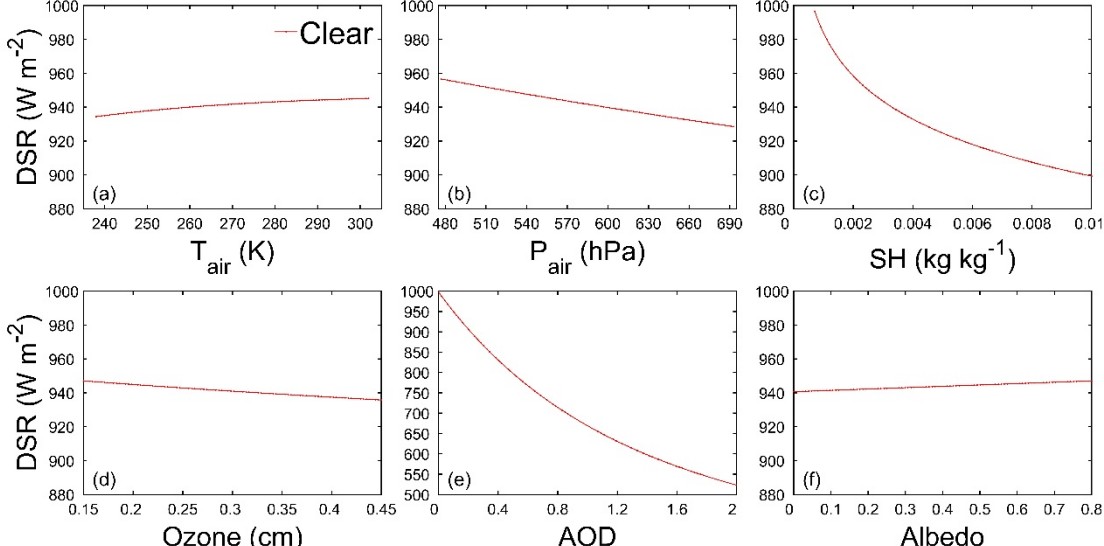

**Figure 9.** Sensitivity of DSR to (a) air temperature $T_{air}$, (b) air pressure $P_{air}$, (c) specific humidity SH, (d) ozone
layer thickness, (e) aerosol optical depth AOD and (f) surface albedo under clear sky conditions.

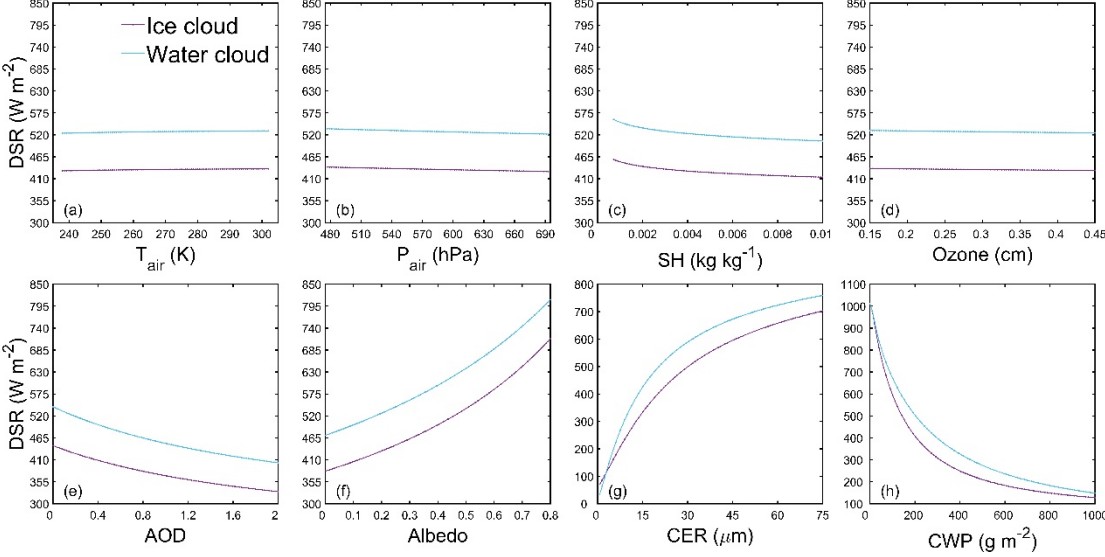

**Figure 10.** Sensitivity of DSR to (a) air temperature $T_{air}$, (b) air pressure $P_{air}$, (c) specific humidity SH, (d) ozone
layer thickness, (e) aerosol optical depth AOD, (f) surface albedo, (g) cloud effective radius CER and (h) cloud
water path CWP under cloudy sky conditions for ice clouds (purple line) and water clouds (blue line).

Moreover, the fluctuating range of input variables within one standard deviation (1σ) and the
induced DSR fluctuation under different sky conditions are summarized in Table 6. Under clear sky
conditions, the DSR is highly sensitive to AOD and SH and only slightly sensitive to other input

variables. The AOD and SH within $1\sigma$ correspond to ranges of approximately 0-0.23 and 0.0004-0.0047 kg kg$^{-1}$, respectively, which would lead to DSR fluctuating by approximately 100.6 W m$^{-2}$ and 87.4 W m$^{-2}$, respectively. Other input variables only induce fluctuations in DSR smaller than 15 W m$^{-2}$. Under cloudy sky conditions, the DSR shows significant sensitivity to CWP and CER, moderate sensitivity to albedo, SH and AOD, and slight sensitivity to other input variables. The CWP within the $1\sigma$ range would lead to DSR fluctuating by approximately 768.1 W m$^{-2}$ and 526.7 W m$^{-2}$ for ice clouds and water clouds, respectively. The CER within the $1\sigma$ range would lead to DSR fluctuating by approximately 212.2 W m$^{-2}$ and 202.3 W m$^{-2}$ for ice clouds and water clouds, respectively. The magnitude of DSR fluctuations induced by the remaining input variables is much smaller than that caused by CWP and CER. In addition, the sensitivity of DSR to albedo is higher under cloudy sky conditions than under clear sky conditions, while the sensitivity of DSR to AOD and SH is lower under cloudy sky conditions than under clear sky conditions.

**Table 6.** Fluctuating range of input variables within one standard deviation ($1\sigma$) and the induced DSR fluctuation under clear sky and cloudy sky conditions.

| Variables | Clear | | Ice cloud | | Water cloud | |
|---|---|---|---|---|---|---|
| | Ranges of variables within $1\sigma$ | DSR fluctuation range (W m$^{-2}$) | Ranges of variables within $1\sigma$ | DSR fluctuation range (W m$^{-2}$) | Ranges of variables within $1\sigma$ | DSR fluctuation range (W m$^{-2}$) |
| $T_{air}$ (K) | 264-282 | 2.6 | 263-282 | 1.3 | 271-288 | 1.1 |
| $P_{air}$ (hPa) | 530-622 | -12.0 | 537-633 | -4.9 | 550-646 | -5.7 |
| SH (kg kg$^{-1}$) | 0.0004-0.0047 | -87.4 | 0.0006-0.0059 | -38.0 | 0.0035-0.0083 | -17.34 |
| Ozone (cm) | 0.25-0.28 | -1.3 | 0.25-0.30 | -0.7 | 0.25-0.28 | -0.6 |
| AOD | 0-0.23 | -100.6 | 0.03-0.21 | -19.7 | 0.06-0.23 | -21.1 |
| Albedo | 0.09-0.32 | 1.8 | 0.08-0.35 | 82.9 | 0.06-0.29 | 65.7 |
| CER (μm) | - | - | 16.7-39.8 | 212.2 | 9.3-21.4 | 202.3 |
| CWP (g m$^{-2}$) | - | - | 0-409.6 | -768.1 | 29.8-351.1 | -526.7 |

In general, the inputs of cloud parameters CWP and CER are crucial variables, and their sensitivities are consistently high. AOD, surface albedo and SH are of secondary importance, with moderate sensitivity. AOD and surface albedo are more sensitive to DSR estimation than SH. Tair, Pair and ozone layer thickness only have a slight sensitivity to DSR estimation, in which ozone layer thickness is the least sensitive. The sensitivity test results indicate that the uncertainties in the input data of cloud parameters, aerosol parameters, surface albedo, and water vapour content are important error sources in the estimation of DSR (Huang et al., 2020; Letu et al., 2020).

**5 Summary**

Various satellite-based methods for estimating DSR have been developed during the past few decades, but some of them rarely operate effectively over the TP due to its complex terrain, high elevation, and unique climatology. Current surface radiation products ignore the influence of topographic variability on the DSR by simply assuming that the surface is horizontal and uniform, resulting in unreliable estimations in rugged regions. Due to the complexity and heterogeneity of the underlying surface of the TP, it is indispensable to consider the topographic variability in the process of DSR estimation. However, few models take the terrain effect into account on the large spatial scale of the whole TP. Unlike aerosol scattering and Rayleigh scattering, multiple scattering plays an important role in DSR attenuations caused by clouds. However, radiative extinctions due to cloud multiscattering tend to be ignored in existing DSR estimation methods under cloudy-sky conditions.

Thus, an improved parameterization scheme for deriving DSR over the TP under all-sky conditions is proposed in this paper. Based on meteorological forcing data and satellite data, the effects caused by ozone, aerosol, water vapor, Rayleigh scattering, permanent gas, cloud single scattering, cloud multiple scattering and topography are comprehensively considered in the improved parameterization scheme. The estimated DSR was validated against in situ observations collected at 12 stations over the TP, which cover a variety of elevations, climates, and land cover types. The validation results on different temporal scales show that the derived DSR based on the developed scheme is in good agreement with ground measurements. By comparing with existing widely used DSR products based on the same in situ observations, the derived DSR of this study performed better with the smallest RMSE, the lowest absolute value MB and the comparable R values on different spatiotemporal scales. Furthermore, the derived DSR of this study can capture the temporal variation characteristics as revealed by in situ observations. The proposed methodology also provided reasonable spatial distribution patterns. Specifically, this method demonstrated its superiority in characterizing more details and high dynamics of the spatial pattern of DSR due to its higher resolution (1 km) and terrain correction. In addition, the differences in the verification results and spatial distribution of different DSR products also prove that there are still great uncertainties in current DSR products over the TP.

It should be noted that there are still some discrepancies for estimated DSR. Several reasons may contribute to these discrepancies. First, the accuracy of the parameterization method depends on the

accuracy of the input data to some extent, such as cloud and aerosol information. At present, the

545 inhomogeneity of the horizontal and vertical directions of clouds in nature cannot be fully reflected

from the plan-parallel assumption, which is used for most cloud physical parameter inversions (Letu et

al., 2020). The defects will lead to uncertainties in cloud parameters. For the input atmospheric

parameters, the retrieval of AOD is quite challenging. The current popular "dark target" algorithm

cannot deal well with AOD retrievals on bright surfaces, such as snow/ice cover. Some studies have

550 shown that MODIS AOD products have high uncertainties in the TP compared with other regions

(Wang et al., 2007; Xu et al., 2015). Second, there are many snow/ice covers in the TP, while snow/ice

and clouds are hard to distinguish due to their similar reflective optical characteristics in many spectral

regions. The ground radiation field becomes extremely complex when the surface is covered by

snow/ice. These factors make it still a very challenging task to estimate the DSR on snow/ice cover

thus far, especially under cloudy-sky conditions. Finally, kilometer-level satellite-based DSR is

susceptible to the 3D radiative effects of clouds. It is difficult to tackle the 3D variability of clouds in

DSR retrieval algorithms, especially for instantaneous DSR (Huang et al., 2019). Furthermore, because

convective clouds are abundant and easily lead to precipitation over the TP (Fu et al., 2020), the 3D

effect of clouds may be more difficult to address on the TP.

The improved parameterization scheme can provide an independent reference for surface radiation

budget and land–atmosphere interaction studies over the TP. In this study, topographic effects are

coupled in the DSR parameterization scheme by taking shading and terrain reflections into account.

Sky view factor is also an important factor for DSR in mountainous areas (Ma et al., 2023). Further

improvements may be achieved by introducing the sky view factor into the parameterization scheme.

It's still a great challenge to evaluate DSR products over mountainous areas. Currently, it is difficult to

do fully evaluations for this complex topography due to lack of in situ measurements on different

aspect and slopes over the TP (Yan et al., 2020; Ma et al., 2023). Additionally, the generating of daily

shortwave radiation datasets remains a challenge. New-generation geostationary satellites with higher

temporal and spectral resolutions, such as FengYun-4 and Himawari-8, have been launched

successfully (Bessho et al., 2016; Guo et al., 2017). This provides an opportunity to obtain hourly and

daily DSR. Moreover, this allows us to further extend this method to obtain more details of surface

radiation components over the TP in the future.

## Appendix A

$$\tau_{b,clr} = max(0, \tau_a \tau_{oz} \tau_g \tau_w - 0.013) \cos\theta, \tag{A1}$$

$$\tau_{d,clr} = max(0, \tau_{oz} \tau_g \tau_w (1 - \tau_r \tau_a) + 0.013) \frac{(cos\,s)^2}{2\,sin\,\alpha}, \tag{A2}$$

$$\tau_{r,clr} = \rho_g (0.271 + 0.706 max(0, \tau_a \tau_{oz} \tau_g \tau_w - 0.013)) \frac{(sin\,s)^2}{2\,sin\,\alpha}, \tag{A3}$$

where $\tau_a$, $\tau_{oz}$, $\tau_g$, $\tau_w$ and $\tau_r$ refer to the broadband radiative transmittance for ozone aerosol extinction (aerosol scattering and absorption), ozone absorption, permanent gas absorption, water vapor absorption, and Rayleigh scattering, respectively. The above transmittances, $\tau_a$, $\tau_{oz}$, $\tau_g$, $\tau_w$, and $\tau_r$, were obtained primarily by the parameterizations of Yang et al. (2006a).

$$\tau_{b,cld} = \tau_{oz} \tau_w \tau_g \tau_r \tau_a \tau_{cld} \cos\theta, \tag{A4}$$

$$\tau_{d,cld} = (\tau_{d,cld}^{r,ms} + \tau_{d,cld}^{a,ms} + \tau_{d,cld}^{ss,ms}) \frac{(cos\,s)^2}{2\,sin\,\alpha}, \tag{A5}$$

$$\tau_{d,cld}^{r,ms} = 0.5 \tau_{oz} \tau_w \tau_g \tau_{aa} (1 - \tau_r) \tau_{cld,a} \tau_{cld,ms}, \tag{A6}$$

$$\tau_{d,cld}^{a,ms} = f_{aer}(\mu) \tau_{oz} \tau_w \tau_g \tau_{aa} \tau_r (1 - \tau_{as}) \tau_{cld,a} \tau_{cld,ms}, \tag{A7}$$

$$\tau_{d,cld}^{ss,ms} = \tau_{oz} \tau_w \tau_g \tau_a \tau_r \tau_{cld,a} (1 - \tau_{cld,ss}) \tau_{cld,ms}, \tag{A8}$$

$$\tau_{r,cld} = \rho_g (0.271 + 0.706 \tau_{oz} \tau_w \tau_g \tau_r \tau_a \tau_{cld}) \frac{(sin\,s)^2}{2\,sin\,\alpha}, \tag{A9}$$

where $\tau_{cld}$, $\tau_{cld,a}$ and $\tau_{cld,ss}$ refer to the broadband radiative transmittance, broadband radiative absorption transmittance and broadband radiative scattering transmittance caused by cloud single-scattering actions, respectively. $\tau_{aa}$, $\tau_{as}$ and $\tau_{cld,ms}$ refer to the broadband radiative transmittance for aerosol absorption, aerosol scattering and cloud radiation multiple actions, respectively.

$\mu$ is the cosine of the solar zenith angle, and $f_{aer}(\mu)$ is the aerosol forward scattering fraction, which is parameterized as

$$f_{aer}(\mu) = 0.364 + 0.632\mu - 0.245\mu^2, \tag{A10}$$

$\tau_{cld}$, $\tau_{cld,a}$, $\tau_{cld,ss}$, $\tau_{cld,ms}$, $\tau_{aa}$, and $\tau_{as}$ can be described as follows:

$$\tau_{cld} = exp(-aCWP/\mu CER), \tag{A11}$$

$$\tau_{cld,a} = exp(-bCWP/\mu CER), \tag{A12}$$

$$\tau_{cld,ss} = exp(-c_1\mu CWP^{C_2}/(\mu^{C_2} + c_3 CWP^{C_2})), \tag{A13}$$

$$\tau_{cld,ms} = exp(\frac{-CWP/CER}{d_1+d_2CWP/CER+d_3\sqrt[2]{CWP/CER}}), \tag{A14}$$

$\quad \tau_{aa} = \tau_a^{(1-\omega_a)}, \tag{A15}$

$$\tau_{as} = \tau_a^{\omega_a}, \tag{A16}$$

The atmosphere hemispherical albedo $\rho_{a,cld}$ is parameterized as:

$$\rho_{a,cld} = 0.086 + \frac{CWP/CER}{e_1+e_2CWP/CER+e_3\sqrt[2]{CWP/CER}}, \tag{A17}$$

where the coefficients $(a, b, c_1, c_2, c_3, d_1, d_2, d_3, e_1, e_2, e_3)$ for different types of clouds can be found in

the study by Huang et al. (2018). $\omega_a$ is the aerosol single-scattering albedo, and its value depends on

the type of aerosol (Levy et al., 2007; Huang et al., 2020).

Here, we assume that ozone absorption and air molecule scattering both take place above clouds

(Qin et al., 2015; Tang et al., 2016; Huang et al., 2018). $\tau_{d,cld}^{r,ms}$ and $\tau_{d,cld}^{a,ms}$ can represent the part of

diffuse radiation (caused by Rayleigh scattering and aerosol scattering, respectively) that finally

reaches the surface after cloud multiscattering. $\tau_{d,cld}^{ss,ms}$ can represent the part of diffuse radiation

(caused by cloud single scattering) that finally reaches the surface after cloud multiscattering.

The topographic effects are taken into account in DSR estimation parameterization schemes by

the solar zenith angle $\theta$, the solar altitude angle $\alpha$ and the tilt angle of the surface (slope) $s$.

According to this knowledge, Chen et al. (2013) provided a scheme that can be applied in mountainous

areas based on high-resolution DEM datasets.

$$sin\,\alpha = sin\,L\,sin\,\delta_s + cos\,L\,cos\,\delta_s\,cos\,h_s, \tag{A18}$$

$cos\,\theta = sin\,L\,sin\,\delta_s\,cos\,s - cos\,L\,sin\,\delta_s\,sin\,s\,cos\,\gamma$

$+ cos\,L\,cos\,\delta_s\,cos\,s\,cos\,h_s + sin\,L\,cos\,\delta_s\,sin\,s\,cos\,\gamma\,cos\,h_s$

$$+ sin\,L\,cos\,\delta_s\,sin\,s\,sin\,h_s, \tag{A19}$$

where $L$ is latitude. $\delta_s$ is the declination of the earth. $h_s$ is the hour angle. $\gamma$ is the surface aspect

angle.

The BSA and WSA are the surface albedos under the condition of complete direct and diffuse

solar radiation, not the actual surface albedo. According to Pinty et al. (2005) and Stokes and Schwartz (1994), the actual surface albedo can be obtained by

$$r = 0.122 + 0.85exp(-4.8\mu), \tag{A20}$$

$$\rho_g = rBSA + (1-r)WSA, \tag{A21}$$

The precipitable water w (cm) is estimated from relative humidity RH (%) and air temperature $T_{air}$ (K) by a semiempirical formula (Yang et al., 2006; Yang et al., 2010):

$$w = 0.00493RHT_{air}^{-1}exp(26.23 - 5416T_{air}^{-1}), \tag{A22}$$

**Data availability**

The in situ measurements used in this study were obtained from the National Tibetan Plateau Data Center (http://data.tpdc.ac.cn) and National Cryosphere Desert Data Center (http://www.ncdc.ac.cn). The MODIS products we used can be freely downloaded via the NASA website (https://modis.gsfc.nasa.gov/).

**Author contributions.**

PL and ZL designed and implemented the study. PL prepared the manuscript with help from ZL, YM and YF. MC and XW contributed to analysis of the data. YQ and ZW collected the in situ data. All commented on the paper.

**Acknowledgments**

This work was funded by the Second Tibetan Plateau Scientific Expedition and Research (STEP) Program (Grant No. 2019QZKK0103); The Chinese Academy of Sciences (Grant No. XDA20060101); National Natural Science Foundation of China (Grant No. 41875031, 91837208, 41522501, 41275028) and CLIMATE-Pan-TPE in the framework of the ESA-MOST Dragon 5 Programme (Grant ID 58516).

**Competing interests.**

The authors declare that they have no conflict of interest.

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
