# Peer review of "Estimation of 1 km downwelling shortwave radiation over the Tibetan Plateau under all-sky conditions"

_Atmospheric Chemistry and Physics, 2022_

## Referee Comment (RC1)

This manuscript improved the parameterization scheme in terms of topography and cloud multi-scattering and generated a 1-km DSR product over the Tibetan Plateau. The topic is interesting and the DSR generation over Tibetan Plateau has been a great challenge over years. Overall, this study achieved high-accuracy DSR estimation over Tibetan Plateau, yet some details need to be clarified. I hope the authors could conduct some of my suggestions and comments, which may help to make this study much better. Here are some major concerns:

- In L134, as far as I know, MCD18A1 now offers a 1 km daily DSR. Letu et al. (2022) also generated a DSR product with topographic consideration at 10-min and 0.05° over East Asia-Pacific. I would like to know what is the advantage of the generated DSR in this study compared with them. A comparison with them would enhance the superiority of this study. See Letu et al. (2022) A New Benchmark for Surface Radiation Products over the East Asia–Pacific Region Retrieved from the Himawari-8/AHI Next-Generation Geostationary Satellite.
- 2. For Table 1, this method relied on atmospheric products, so a sensitivity analysis could be conducted to show the reliability and possible issues of the method. For example, introduce 10% random errors (this error depends on the performances of atmospheric products) to the input, and see how much the estimation result change.
- 3. In Table 2, could you please offer the slope and aspect of the stations? Here is a big concern that most stations are located in relatively flat areas in mountains, and as Figure 7 showed that DSR varied greatly with different terrain conditions, the 1 km\* 1 km pixel has sub-topography, and the ground-measured shortwave flux could not represent the pixel-scale DSR in mountains. I think it is a great challenge to evaluate DSR products over mountainous areas. See Yan et al. (2020) An Operational Method for Validating the Downward Shortwave Radiation Over Rugged Terrains.
- 4. In Table 3, could you provide the sample size of each site at each timescale? I find that the sample size in Figure 3 was not large (N=155 on monthly scale), so I am afraid the different performances were attributed to the sample size in the validation.
- 5. In L525, did you consider the shelters from surrounding mountains to the target pixel's DSR? e.g., sky view factor and shadow. You can see that sky view factor matters, especially under cloudy sky: Ma et al. (2023) Estimation of fine spatial resolution all-sky surface net shortwave radiation over mountainous terrain from Landsat 8 and Sentinel-2 data

Here are some minor concerns:

- 1. Could you be specific about what is high resolution (i.e., maybe < 5 km?) and what is coarse resolution in this study? Is the estimated DSR at an instantaneous scale? I am curious how you upscaled it to ten-day and monthly timescales for evaluation.
- 2. In L62, I would like to know why the DSR could exceed the solar constant.
- 3. In L116, the references in the 1990s are old, now many all-sky DSR products have been released.
- 4. In L120, you mentioned many parameterization schemes did not consider the DSR attenuation caused by clouds carefully enough. This sentence is subjective, and could you please give some references? The next sentence should be improved, too.

5. In L289, these DSR products need to be introduced in the Data Section. I think CERES\_SYN\_1h should be CERES SYN1deg-1 Hour, am I right?

---

## Author Comment (AC1)

Dear Reviewers:

We greatly appreciate the reviewer's efforts in reviewing our manuscript. All the comments are very valuable for improving our manuscript. Below are the detailed point-by-point responses to the review comments. For clarity, the referees' comments are listed in black italics, and our responses and changes in the manuscript are shown in blue. We also mention where we made necessary changes in the revised manuscript by indicating page and line numbers in our responses. Please find our point-by-point responses below.

**Response to Reviewer #3**

*This manuscript improved the parameterization scheme in terms of topography and cloud multi-scattering and generated a 1-km DSR product over the Tibetan Plateau. The topic is interesting and the DSR generation over Tibetan Plateau has been a great challenge over years. Overall, this study achieved high-accuracy DSR estimation over Tibetan Plateau, yet some details need to be clarified. I hope the authors could conduct some of my suggestions and comments, which may help to make this study much better.*

**Author Response:** Thank you very much for your valuable suggestions and thoughtful instructions. All comments were helpful for improving our manuscript. We carefully revised the manuscript and made the following point-by-point revisions according to your suggestions.

**Major concerns:**

1. *In L134, as far as I know, MCD18A1 now offers a 1 km daily DSR. Letu et al. (2022) also generated a DSR product with topographic consideration at 10-min and 0.05° over East Asia-Pacific. I would like to know what is the advantage of the generated DSR in this study compared with them. A comparison with them would enhance the superiority of this study. See Letu et al. (2022) A New Benchmark for Surface Radiation Products over the East Asia–Pacific Region Retrieved from the Himawari-8/AHI Next-Generation Geostationary Satellite.*

**Author Response:** Thank you for this comment. The comparison results with MCD18A1 and Letu et al. (2020) have been added to Table 4. It should be noted that there are many missing values over the TP in MCD18A1. The same phenomenon was

found in Li et al. (2022). Fig. 1 will not be added to the revised manuscript, as it is only shown here to illustrate that there are many missing values over the TP in MCD18A1. To avoid the potential uncertainty caused by different sample sizes, the results given in Table 4 are of the same sample size.

The DSR product generated by Letu et al. (2022) (short for "H-8_EAP") is based on the Himawari-8/AHI satellite at a 10-min temporal scale and 5-km spatial scale over the East Asia-Pacific. The earliest time covered by this product was 2016. At present, the latest in situ data in this study are in 2016. In addition, the Himawari-8 satellite cannot observe the western part of the TP. The spatial range of the product cannot cover the entire TP. Therefore, six stations (BJ, QOMS, SETORS, NAMORS, NLGS and NLTS) in 2016 are selected to compare our product with H-8_EAP.

The RMSEs of MCD18A1 at three temporal scales are 233.47, 147.04 and 130.24 W m$^{-2}$, respectively. The MBs of MCD18A1 at three temporal scales are -76.43, -74.60 and -74.17 W m$^{-2}$, respectively. The estimates of this study show smaller RMSEs (152.13, 77.24 and 63.79 W m$^{-2}$) and lower absolute value MBs (5.23, 7.35 and 63.79 W m$^{-2}$). For H-8_EAP, the RMSEs at three temporal scales are 197.89, 140.67 and 125.70 W m$^{-2}$, respectively. The MBs at three temporal scales are -52.47, -57.07 and -62.74 W m$^{-2}$, respectively. The estimates of this study show smaller RMSEs (140.54, 82.67 and 71.48 W m$^{-2}$) and lower absolute value MBs (23.64, 21.54 and 14.97 W m$^{-2}$).

Table 4. Comparison with existing DSR products on different timescales in terms of accuracy.

| Product name | Instantaneous timescale | | | Ten-day timescale | | | Monthly timescale | | | Spatial resolution |
|---|---|---|---|---|---|---|---|---|---|---|
| | RMSE (W m$^{-2}$) | MB (W m$^{-2}$) | R | RMSE (W m$^{-2}$) | MB (W m$^{-2}$) | R | RMSE (W m$^{-2}$) | MB (W m$^{-2}$) | R | |
| MCD18A1 | 233.47 | -76.43 | 0.60 | 147.04 | -74.60 | 0.72 | 130.24 | -74.17 | 0.74 | 1 km |
| This study | 152.13 | 5.23 | 0.72 | 77.24 | 7.35 | 0.82 | 63.79 | 7.25 | 0.84 | |
| H-8_EAP | 197.89 | -52.47 | 0.66 | 140.67 | -57.07 | 0.67 | 125.70 | -62.74 | 0.73 | 5 km |
| This study | 140.54 | 23.64 | 0.77 | 82.67 | 21.54 | 0.78 | 71.48 | 14.97 | 0.81 | |
| ERA5 | 165.67 | -20.59 | 0.65 | 88.06 | -21.44 | 0.82 | 74.19 | -21.06 | 0.86 | 25 km |
| This study | 135.11 | 15.67 | 0.77 | 75.01 | 15.24 | 0.83 | 67.12 | 15.75 | 0.83 | |
| CERES_SYN_1h | 146.64 | -46.70 | 0.75 | 84.27 | -47.93 | 0.86 | 73.25 | -47.53 | 0.89 | 100 km |
| CERES_SYN_3h | 160.50 | -78.30 | 0.74 | 107.13 | -79.48 | 0.85 | 98.67 | -79.06 | 0.88 | |
| GEWEX_SRB | 194.45 | -118.56 | 0.68 | 143.68 | -119.71 | 0.80 | 135.54 | -119.21 | 0.83 | |
| This study | 132.84 | 2.79 | 0.77 | 70.84 | 2.18 | 0.84 | 61.33 | 2.70 | 0.85 | |

[Figure]

**Figure 1.** Comparison between the estimated instantaneous DSR and in situ measurements for (a) this study and (b) MCD18A1. N indicates the number of points. The legend with different colors denotes the twelve stations involved in the validation. The units of RMSE, MB and MAE are W m⁻².

Considering the available years of different DSR products, as well as the integrity and temporal continuity of in situ data, the MCD18A1 product was added to the comparison. The corresponding Figure 4 has been updated in the revised manuscript. Compared with other products, the DSR derived in this study is more consistent with the in situ observations at each station, and all show similar temporal change trends.

[Figure]

**Figure 4.** Intercomparison of time series of DSR among MCD18A1, ERA5, CERES_SYN_1 h, and this study at (a) BJ, (b) D105, (c) NPAM, (d) SETORS, (e) QOMS, (f) MAWORS, (g) NADORS, (h) NAMORS, (i) NLGS, (j) NLTS, (k) XDT, and (l) TGL stations on a ten-day timescale. The circle denotes in situ data.

Relevant statements have been updated in the revised manuscript as follows. (P14, L301-L312; P15, L323-L331; P16, L345-L348; P16, L353-354).

'Among these products, there are remotely sensed and reanalysis DSR products, namely, Clouds and the Earth's Radiant Energy System Synoptic (CERES_SYN) surface fluxes (Loeb et al., 2013), Global Energy and Water Exchanges Surface Radiation Budget (GEWEX_SRB) datasets (Zhang et al., 2014), MODIS DSR product (MCD18A1) (Wang et al., 2020) and the fifth generation reanalysis (ERA5) from the European Centre for Medium-Range Weather Forecasts (ECMWF) (Hans et al., 2019). In addition, Letu et al. (2022) produced a high-resolution (5 km, 10 min) DSR dataset (short for "H-8_EAP" in our study) under all-sky conditions from 2016 to 2020 in the East Asia–Pacific region based on the next-generation geostationary satellite Himawari-8/AHI, which was also selected for comparison. At present, the latest in situ data in this study are in 2016, and the Himawari-8 satellite cannot observe the western part of the TP. Therefore, six stations (BJ, QOMS, SETORS, NAMORS, NLGS and NLTS) in 2016 are selected for comparison with the H-8_EAP DSR dataset.' (P14, L301-L312)

'As summarized in Table 4, the RMSE range of these DSR products is approximately 150~230 W $m^{-2}$ at the instantaneous scale. At the ten-day scale, the RMSE range is approximately 80~150 W $m^{-2}$. At the monthly scale, the RMSE range is approximately 70~130 W $m^{-2}$. The MB range of these DSR products is -120 ~ -20 W $m^{-2}$ at three temporal scales. These large spans of RMSE and MB indicate that the current DSR products still have great uncertainties over the TP. The RMSE ranges of this study at three temporal scales are 132~152, 70~82 and 61~71 W $m^{-2}$. The MB range of this study is 3 ~ 24 W $m^{-2}$ at three temporal scales. The estimates of this study show a smaller RMSE, lower absolute value MB and comparable R values at the corresponding spatial and temporal scales. This means that the derived DSR based on the proposed method performs better than other DSR products over the TP.' (P15, L323-L331)

'DSR products with relatively high accuracy, which correspond to three spatial resolutions of 1 km, 25 km and 100 km, are selected for comparison with the estimated DSR in this study in terms of temporal variation characteristics (Fig. 4). The time series of MCD18A1 at NAMORS and NLGS stations are not displayed because there are many missing values in MCD18A1 at these two stations.' (P16,

L345-L348)

'The dynamic range (defined as the difference between the maximum and the minimum in a year) of MCD18A1 is the largest, while ERA5, CERES_SYN_1 h and this study show similar dynamic ranges.' (P16, L353-354)

The related reference has been added in the revised manuscript as follows (P32, L718; P37, L855):

Letu, H., Nakajima, T. Y., Wang, T., Shang, H., Ma, R., Yang, K., Baran, A. J., Riedi, J., Ishimoto, H., Yoshida, M., Shi, C., Khatri, P., Du, Y., Chen, L., and Shi, J.: A new benchmark for surface radiation products over the East Asia–Pacific region retrieved from the Himawari-8/AHI next-generation geostationary satellite, Bulletin of the American Meteorological Society, 103, E873-E888, 10.1175/bams-d-20-0148.1, 2022. (P32, L718)

Wang, D., Liang, S., Zhang, Y., Gao, X., Brown, M. G. L., and Jia, A.: A new set of MODIS land products (MCD18): Downward shortwave radiation and photosynthetically active radiation, Remote Sensing, 12, 10.3390/rs12010168, 2020. (P37, L855)

2. *For Table 1, this method relied on atmospheric products, so a sensitivity analysis could be conducted to show the reliability and possible issues of the method. For example, introduce 10% random errors (this error depends on the performances of atmospheric products) to the input, and see how much the estimation result change.*

**Author Response:** Thank you for this comment. The accuracy of the parameterization scheme depends on the quality of the input data to some extent. To further understand the effect of uncertainties in input variables on the accuracy of the DSR retrieval scheme, sensitivity analysis of the DSR to input variables is conducted (Fig. 9 and Fig. 10). As shown in Fig. 8, three points located in the west, north central, and southeast of the TP are randomly selected for sensitivity tests. The average of each input variable (including air temperature Tair, air pressure Pair, specific humidity SH, ozone layer thickness, aerosol optical depth AOD, surface albedo, cloud effective radius CER and cloud water path CWP) for three randomly selected points is selected as the default value.

[revised manuscript text omitted]

The above information has been added to the revised manuscript (P22, L459-P24,

L513).

3. *In Table 2, could you please offer the slope and aspect of the stations? Here is a big concern that most stations are located in relatively flat areas in mountains, and as Figure 7 showed that DSR varied greatly with different terrain conditions, the 1 km\* 1 km pixel has sub-topography, and the ground-measured shortwave flux could not represent the pixel-scale DSR in mountains. I think it is a great challenge to evaluate DSR products over mountainous areas. See Yan et al. (2020) An Operational Method for Validating the Downward Shortwave Radiation Over Rugged Terrains.*

**Author Response:** Thank you for this comment. The slope and aspect of the stations are listed in the Table 1.

Table 1. The slope (unit: degree, 0-90) and aspect (unit: degree, change between -180 and 180 from north to west, south and north in anticlockwise direction) information of the station.

| Site | Slope(° from horizontal) | Aspect(° from north) |
|------|--------------------------|----------------------|
| BJ | 0.01 | 0.79 |
| D105 | 0.03 | -1.81 |
| NPAM | 0.01 | 2.55 |
| QOMS | 0.09 | 2.00 |
| SETORS | 0.18 | 0.93 |
| MAWORS | 0.01 | -0.30 |
| NADORS | 0.01 | -2.25 |
| NAMORS | 0.02 | 2.69 |
| NLGS | 0.00 | 2.55 |
| NLTS | 0.00 | 2.15 |
| XDT | 0.05 | 0.29 |
| TGL | 0.04 | -2.24 |

According to the slope and aspect calculated based on DEM data, it can be considered that most stations are located on flat surfaces. It is true that evaluating DSR products over mountainous areas is a great challenge. Yan et al. (2020) proposed a methodology to validate DSR in mountainous areas. The premise of this method is that more than three stations are required on different slopes within kilometer-scale areas with rugged terrain (Yan et al. 2020). Currently, there are no such measurements on the TP, which makes it difficult to validate DSR, as the reviewer mentioned. For well-known reasons, setting up such measurements over the TP with extremely high altitude and harsh climatic environment is very costly and difficult.

4. *In Table 3, could you provide the sample size of each site at each timescale? I find that the sample size in Figure 3 was not large (N=155 on monthly scale), so I am afraid the different performances were attributed to the sample size in the validation.*

**Author Response:** Thank you for this comment. We have added the sample size of each site at each timescale to Table 3. At the same temporal scale, the sample size for each site does not differ significantly. Therefore, we believe that the sample size has little impact on the final validation result.

**Table 3.** Summary statistics of the validation results for each station on different timescales.

| Site | Instantaneous timescale | | | | Ten-day timescale | | | | Monthly timescale | | | | CCD |
|---|---|---|---|---|---|---|---|---|---|---|---|---|---|
| | RMSE (W m⁻²) | MB (W m⁻²) | R | N | RMSE (W m⁻²) | MB (W m⁻²) | R | N | RMSE (W m⁻²) | MB (W m⁻²) | R | N | |
| BJ | 179.44 | 11.41 | 0.66 | 359 | 66.20 | 13.91 | 0.84 | 36 | 56.01 | 14.54 | 0.81 | 12 | 49.58% |
| D105 | 162.87 | 32.47 | 0.67 | 359 | 76.69 | 33.23 | 0.73 | 36 | 67.43 | 33.80 | 0.73 | 12 | 54.02% |
| NPAM | 177.57 | -3.90 | 0.63 | 358 | 67.63 | -4.28 | 0.82 | 36 | 51.90 | -3.75 | 0.82 | 12 | 53.46% |
| QOMS | 112.33 | 5.04 | 0.74 | 689 | 56.49 | 6.38 | 0.90 | 69 | 49.76 | 6.41 | 0.91 | 23 | 19.83% |
| SETORS | 183.33 | -49.51 | 0.67 | 302 | 94.17 | -49.48 | 0.67 | 33 | 64.89 | -44.04 | 0.74 | 12 | 72.85% |
| MAWORS | 167.41 | 28.51 | 0.71 | 350 | 83.27 | 27.08 | 0.90 | 36 | 72.94 | 27.32 | 0.92 | 12 | 55.62% |
| NADORS | 129.88 | 19.48 | 0.78 | 318 | 66.20 | 17.59 | 0.89 | 36 | 58.30 | 18.20 | 0.90 | 12 | 35.07% |
| NAMORS | 150.62 | 18.30 | 0.72 | 342 | 65.60 | 13.66 | 0.88 | 36 | 55.92 | 13.42 | 0.89 | 12 | 40.27% |
| NLGS | 141.53 | 11.26 | 0.77 | 365 | 66.51 | 10.81 | 0.81 | 36 | 56.48 | 11.02 | 0.80 | 12 | 46.58% |
| NLTS | 136.29 | 24.63 | 0.79 | 360 | 62.80 | 22.01 | 0.86 | 36 | 51.55 | 23.81 | 0.87 | 12 | 59.45% |
| XDT | 183.08 | 17.84 | 0.63 | 365 | 81.41 | 17.95 | 0.72 | 36 | 70.48 | 18.02 | 0.70 | 12 | 51.23% |
| TGL | 188.98 | -46.64 | 0.58 | 365 | 97.70 | -46.52 | 0.72 | 36 | 87.80 | -46.92 | 0.66 | 12 | 45.63% |

5. *In L525, did you consider the shelters from surrounding mountains to the target pixel's DSR? e.g., sky view factor and shadow. You can see that sky view factor matters, especially under cloudy sky: Ma et al. (2023) Estimation of fine spatial resolution all-sky surface net shortwave radiation over mountainous terrain from Landsat 8 and Sentinel-2 data.*

**Author Response:** Thank you for this comment. This parameterization scheme for calculating the DSR was improved by considering variations in the slope and azimuth of the land surface, as well as the terrain shadow in mountainous areas. The sky view factor was not taken into account in this study. Nevertheless, the radiation model over rugged terrains used in this study has already been validated in former studies. Therefore, we think it is feasible to use this model for the TP. Your suggestion of

introducing the sky view factor into the scheme may further improve the accuracy of the estimation. However, calculating solar radiation over rugged terrain is a complex task, as depicted in Ma et al. (2023), and more detailed considerations can be carried out in future work.

**Here are some minor concerns:**

1. *Could you be specific about what is high resolution (i.e., maybe < 5 km?) and what is coarse resolution in this study? Is the estimated DSR at an instantaneous scale? I am curious how you upscaled it to ten-day and monthly timescales for evaluation.*

**Author Response:** High resolution and coarse resolution are relative concepts that usually depend on the spatial coverage of the study area. For the TP, a spatial resolution finer than 5 km can be considered high resolution, and a spatial resolution coarser than 10 km can be considered coarse resolution. Since the main input data MODIS is at an instantaneous scale, the estimated DSR is also at an instantaneous scale. It is upscaled to ten-day and monthly timescales via arithmetic average for validation.

2. *In L62, I would like to know why the DSR could exceed the solar constant.*

**Author Response:** Due to its high altitude, low airmass and short path for solar radiation to reach the land surface, the TP receives a large amount of DSR. Coupled with multiple scattering of complex terrain, DSR can be higher than the solar constant. At the same time, if clouds that are conducive to scattering appear, the DSR could be higher than the solar constant due to reflection from clouds.

3. *In L116, the references in the 1990s are old, now many all-sky DSR products have been released.*

**Author Response:** The references have been updated in the revised manuscript. The references are as follows:

Huang, G., Li, Z., Li, X., Liang, S., Yang, K., Wang, D., and Zhang, Y.: Estimating surface solar irradiance from satellites: Past, present, and future perspectives, Remote Sensing of Environment, 233, 10.1016/j.rse.2019.111371, 2019.

Letu, H., Shi, J., Li, M., Wang, T., Shang, H., Lei, Y., Ji, D., Wen, J., Yang, K., and Chen, L.: A review of the estimation of downward surface shortwave radiation based on satellite data: Methods, progress and problems, Science China Earth

Sciences, 63, 774-789, 10.1007/s11430-019-9589-0, 2020.

4.  *In L120, you mentioned many parameterization schemes did not consider the DSR attenuation caused by clouds carefully enough. This sentence is subjective, and could you please give some references? The next sentence should be improved, too.*

**Author Response:** Thank you for this comment. We have modified the sentence and added some references to the revised manuscript. (L120-L123)

'Second, some parameterization schemes did not consider the DSR attenuation caused by clouds carefully enough. Generally, only the single scattering of clouds was considered, and the multiple scattering effect of clouds was ignored (Huang et al., 2018; Huang et al., 2020).'

The references are as follows:

Huang, G., Liang, S., Lu, N., Ma, M., and Wang, D.: Toward a broadband parameterization scheme for estimating surface solar irradiance: Development and preliminary results on MODIS products, Journal of Geophysical Research: Atmospheres, 123, 12,180-112,193, 10.1029/2018jd028905, 2018.

Huang, G., Li, X., Lu, N., Wang, X., and He, T.: A general parameterization scheme for the estimation of incident photosynthetically active radiation under cloudy skies, IEEE Transactions on Geoscience and Remote Sensing, 58, 6255-6265, 10.1109/tgrs.2020.2976103, 2020.

5.  *In L289, these DSR products need to be introduced in the Data Section. I think CERES_SYN_lh should be CERES SYNldeg-l Hour, am I right?*

**Author Response:** Yes, you are right. According to your suggestion, the introduction to the spatiotemporal resolution of different DSR products has been updated in the revised manuscript as follows (P14, L313-L314; P14, L316-L318).

'The spatial resolutions of MCD18A1 and ERA5 are 1 km and 25 km, respectively. CERES_SYN and GEWEX_SRB have a spatial resolution of 100 km.' (P14, L313-L314)

'The temporal resolution of MCD18A1 is instantaneous. GEWEX_SRB has a temporal resolution of 3 hours, and ERA5 has a temporal resolution of 1 hour. CERES_SYN products have two temporal resolutions of 1 hour and 3 hours.' (P14, L316-L318)

---

## Author Comment (AC2)

Dear Reviewers:

We would like to sincerely thank the reviewer for the thoughtful comments and suggestions. All comments and suggestions have been considered carefully and well addressed. For clarity, the referees' comments are listed in black italics, and our responses and changes in the manuscript are shown in blue. We also mention where we made necessary changes in the revised manuscript by indicating page and line numbers in our responses. Please see our responses to your comments and suggestions below.

**Response to Reviewer #1**

*Reliable downwelling shortwave radiation (DSR) estimation over the Tibetan Plateau (TP) is still a challenging scientific issue. This manuscript developed an improved parameterization scheme to obtain all-sky DSR based on satellite data and meteorological forcing data. The topic of the paper is interesting and it's a good fit for the scope of ACP. The whole paper is well organized with clear logic and robust results. The description of the method is clear. Numbers of work are integrated into this paper, and abundant discussions are presented as well. However, there are still some rooms for the improvement.*

**Author Response:** Thank you very much for your positive comments. All comments were helpful for improving our manuscript. We carefully revised the manuscript and made the following point-by-point revisions according to your suggestions.

**Major concerns:**

1. *The spatial resolution of DSR estimated in this paper is 1 km. The spatial resolution of DSR products for comparison is coarser than 10 km. While the scale of the stations normally represents a scale of about less than 1 km. The authors should give some explanation about their scale mismatch problem.*

**Author Response:** Thank you for your suggestion. The scale mismatch problem is an important and difficult problem to solve in the quantitative remote sensing and atmospheric research fields. Some uncertainties can be induced due to the representativeness errors of point-scale measurements. The insufficient spatial representation of point-scale observations can be partly compensated by lowering their temporal resolution (Hakuba et al., 2013; Huang et al., 2016a; Huang et al.,

2016b). Therefore, the DSR estimation results were also validated at ten-day and monthly timescales to minimize this effect. Relevant statements have been described in Section 4.1 (P12, L262-L266) and Section 4.2 (P15, L332-L335).

'Representativeness errors of point-scale measurements can affect the validation results of instantaneous DSR estimations to some extent. The insufficient spatial representation of point-scale observations can be partly compensated by lowering their temporal resolution (Hakuba et al., 2013; Huang et al., 2016b). Therefore, the DSR estimation results were also validated at ten-day and monthly timescales.' (P12, L262-L266)

'In addition, it is noted that the accuracies of all datasets have been appreciably improved with increasing timescale. This is because the 3D radiative transfer effects and complexity of clouds can be significantly reduced and the spatial representativeness of ground-based measurements can be significantly enhanced through temporal averaging (Huang et al., 2016b; Huang et al., 2016a).' (P15, L332-L335)

If we have enough in situ data within a grid scale of 10 km or 25 km, an average or weighted average of the observations can be directly used to reduce some errors caused by scale mismatch. However, for well-known reasons, it is very difficult to carry out such measurements over the TP with its harsh environment and climate conditions.

The references are as follows:

Hakuba, M. Z., Folini, D., Sanchez-Lorenzo, A., and Wild, M.: Spatial representativeness of ground-based solar radiation measurements, Journal of Geophysical Research: Atmospheres, 118, 8585-8597, 10.1002/jgrd.50673, 2013.

Huang, G., Li, X., Ma, M., Li, H., and Huang, C.: High resolution surface radiation products for studies of regional energy, hydrologic and ecological processes over Heihe river basin, northwest China, Agric. For. Meteorol, 230-231, 67-78, 10.1016/j.agrformet.2016.04.007, 2016a.

Huang, G., Li, X., Huang, C., Liu, S., Ma, Y., and Chen, H.: Representativeness errors of point-scale ground-based solar radiation measurements in the validation of remote sensing products, Remote Sensing of Environment, 181, 198-206, 10.1016/j.rse.2016.04.001, 2016b.

2. *As far as I know, three atmospheric conditions (clear-sky, completely*

*cloud-covered and partially cloud-covered) were distinguished based on cloud fraction data in previous study. In this paper, the author used MOD06 cloud product to distinguish cloud sky and clear sky conditions. Will this cause some uncertainties?*

**Author Response:** Thank you for this comment. Since the cloud fraction is calculated from the 1-km resolution cloud product within a 5-km retrieval region, the spatial resolution of the cloud fraction data is 5 km. For the MODIS cloud fraction data, the 5-km geolocation is copied directly from the center MOD06 cloud product pixel in each 5-km area. Therefore, distinguishing different sky conditions is actually based on a 1-km cloud product. We use the cloud phase to distinguish clear sky and cloudy sky conditions. The spatial resolution of cloud phase data is 1 km. The probability of mixing pixels (i.e., partially cloud-covered) is relatively small at the 1-km spatial scale. Unlike land surface parameters, the spatial heterogeneity of atmospheric parameters is much smaller.

3. *The derived DSR was compared with current widely used DSR products in this paper. That's convincing. To our best knowledge, Letu et al. (2022) generated surface radiation products under all-sky conditions from the Himawari-8/AHI Next-Generation Geostationary Satellite. If the derived DSR can be compared with the latest DSR product, this paper may be more appealing. The reference is as follows,*

   *Letu et al., A New Benchmark for Surface Radiation Products over the East Asia–Pacific Region Retrieved from the Himawari-8/AHI Next-Generation Geostationary Satellite, Bulletin of the American Meteorological Society, 103, E873-E888, 10.1175/bams-d-20-0148.1, 2022*

**Author Response:** Thank you for your suggestion. The DSR product generated by Letu et al. (2022) (short for "H-8_EAP") is based on the Himawari-8/AHI satellite at a 10-min temporal scale and 5-km spatial scale over the East Asia-Pacific. The earliest time covered by this product was 2016. At present, the latest in situ data in this study are from 2016. In addition, the Himawari-8 satellite cannot observe the western part of the TP. The spatial range of the product cannot cover the entire TP. Therefore, six stations (BJ, QOMS, SETORS, NAMORS, NLGS and NLTS) in 2016 are selected to compare our product with H-8_EAP. The corresponding content in Table 4 has been updated in the revised manuscript. The RMSEs of H-8_EAP at three temporal scales

are 197.89, 140.67 and 125.70 W m$^{-2}$, respectively. The MBs of H-8_EAP at three temporal scales are -52.47, -57.07 and -62.74 W m$^{-2}$, respectively. The estimates of this study show smaller RMSEs (140.54, 82.67 and 71.48 W m$^{-2}$) and lower absolute value MBs (23.64, 21.54 and 14.97 W m$^{-2}$). Relevant statements have been updated in the revised manuscript. (P14, L306-L312; P15, L323-L331).

'In addition, Letu et al. (2022) produced a high-resolution (5 km, 10 min) DSR dataset (short for "H-8_EAP" in our study) under all-sky conditions from 2016 to 2020 in the East Asia–Pacific region based on the next-generation geostationary satellite Himawari-8/AHI, which was also selected for comparison. At present, the latest in situ data in this study are in 2016, and the Himawari-8 satellite cannot observe the western part of the TP. Therefore, six stations (BJ, QOMS, SETORS, NAMORS, NLGS and NLTS) in 2016 are selected for comparison with the H-8_EAP DSR dataset.' (P14, L306-L312)

'As summarized in Table 4, the RMSE range of these DSR products is approximately 150~230 W m$^{-2}$ at the instantaneous scale. At the ten-day scale, the RMSE range is approximately 80~150 W m$^{-2}$. At the monthly scale, the RMSE range is approximately 70~130 W m$^{-2}$. The MB range of these DSR products is -120 ~ -20 W m$^{-2}$ at three temporal scales. These large spans of RMSE and MB indicate that the current DSR products still have great uncertainties over the TP. The RMSE ranges of this study at three temporal scales are 132~152, 70~82 and 61~71 W m$^{-2}$. The MB range of this study is 3 ~ 24 W m$^{-2}$ at three temporal scales. The estimates of this study show a smaller RMSE, lower absolute value MB and comparable R values at the corresponding spatial and temporal scales. This means that the derived DSR based on the proposed method performs better than other DSR products over the TP.' (P15, L323-L331)

**Table 4.** Comparison with existing DSR products on different timescales in terms of accuracy.

| Product name | Instantaneous timescale | | | Ten-day timescale | | | Monthly timescale | | | Spatial resolution |
|---|---|---|---|---|---|---|---|---|---|---|
| | RMSE (W m$^{-2}$) | MB (W m$^{-2}$) | R | RMSE (W m$^{-2}$) | MB (W m$^{-2}$) | R | RMSE (W m$^{-2}$) | MB (W m$^{-2}$) | R | |
| MCD18A1 | 233.47 | -76.43 | 0.60 | 147.04 | -74.60 | 0.72 | 130.24 | -74.17 | 0.74 | 1 km |
| This study | 152.13 | 5.23 | 0.72 | 77.24 | 7.35 | 0.82 | 63.79 | 7.25 | 0.84 | |
| H-8_EAP | 197.89 | -52.47 | 0.66 | 140.67 | -57.07 | 0.67 | 125.70 | -62.74 | 0.73 | 5 km |
| This study | 140.54 | 23.64 | 0.77 | 82.67 | 21.54 | 0.78 | 71.48 | 14.97 | 0.81 | |
| ERA5 | 165.67 | -20.59 | 0.65 | 88.06 | -21.44 | 0.82 | 74.19 | -21.06 | 0.86 | 25 km |

| | | | | | | | | | |
|---|---|---|---|---|---|---|---|---|---|
| This study | 135.11 | 15.67 | 0.77 | 75.01 | 15.24 | 0.83 | 67.12 | 15.75 | 0.83 |
| CERES_SYN_1h | 146.64 | -46.70 | 0.75 | 84.27 | -47.93 | 0.86 | 73.25 | -47.53 | 0.89 |
| CERES_SYN_3h | 160.50 | -78.30 | 0.74 | 107.13 | -79.48 | 0.85 | 98.67 | -79.06 | 0.88 |
| GEWEX_SRB | 194.45 | -118.56 | 0.68 | 143.68 | -119.71 | 0.80 | 135.54 | -119.21 | 0.83 |
| This study | 132.84 | 2.79 | 0.77 | 70.84 | 2.18 | 0.84 | 61.33 | 2.70 | 0.85 |

100 km

**Minor issues:**

1. *Figure 1: the caption is too brief. The same problems exist in other figures. Please check and modify.*

**Author Response:** Thank you for your suggestion. The legend of the color map indicates the elevation above mean sea level in meters. We have improved the corresponding figure captions in the revised manuscript.

2. *Page 4, L116: The references cited here are too old. Are there any updated references on relevant studies?*

**Author Response:** Thank you for this comment. The references have been updated in the revised manuscript (P4, L116).

'However, since optical remote sensing is greatly affected by clouds, it is still a big challenge to estimate DSR efficiently and accurately under all-sky conditions (Li et al., 1995; Li et al., 1997; Huang et al., 2019; Zhong et al., 2019; Letu et al., 2020).'

3. *L 46: "It plays a decisive role" => "It plays an important role"*

**Author Response:** It has been corrected. (P2, L46)

4. *L 55: "and other major rivers in Asia originate from the TP" => "and most major rivers in Asia originate from the TP"*

**Author Response:** It has been corrected. (P2, L55)

5. *L 57: "an important research object of global and regional energy" => "an important research object for global and regional energy"*

**Author Response:** It has been corrected. (P2, L57)

---

## Author Comment (AC3)

Dear Reviewers:

We would like to sincerely thank you for your insightful and constructive comments, which have been very helpful to us in improving our manuscript. We carefully considered and fully addressed all comments. Below are the detailed point-by-point responses to the review comments. For clarity, the referees' comments are listed in black italics, and our responses and changes in the manuscript are shown in blue. We also mention where we made necessary changes in the revised manuscript by indicating page and line numbers in our responses. Please find below our point-by-point replies to your comments.

**Response to Reviewer #2**

*A smart DSR retrieval scheme was developed and applied over the Tibetan Plateau. The effect of complex terrain in this region was investigated. The research looks interesting and the results look robust.*

**Author Response:** Thank you very much for your positive comments. All comments were helpful for improving our manuscript. We carefully revised the manuscript and made the following point-by-point revisions according to your suggestions.

**Major concerns:**

1.  *It is not clear how to derive water vapor content, whether it is derived from meteorological observations of temperature or relative humidity or directly from MODIS product.*

**Author Response:** Thank you for this comment. The water vapour content was derived based on air temperature and relative humidity, which are from the China Meteorological Forcing Dataset (CMFD). The precipitable water w (cm) is estimated from relative humidity RH (%) and air temperature $T_{air}$ (K) by a semiempirical formula (Yang et al., 2006; Yang et al., 2010):

$$w = 0.00493\text{RH}T_{air}^{-1}\exp(26.23 - 5416T_{air}^{-1})$$

Relevant statements have been added in the revised manuscript. (P28, L618-L620).

  'The precipitable water w (cm) is estimated from relative humidity RH (%) and air temperature $T_{air}$ (K) by a semiempirical formula (Yang et al., 2006; Yang et al., 2010):

$$w = 0.00493 \mathrm{RH} T_{air}^{-1} \exp(26.23 - 5416 T_{air}^{-1}), \hspace{3cm} \text{(A22)'}$$

The references are as follows:

Yang, K., Koike, T., and Ye, B.: Improving estimation of hourly, daily, and monthly solar radiation by importing global data sets, Agric. For. Meteorol, 137, 43-55, 10.1016/j.agrformet.2006.02.001, 2006a.

Yang, K., He, J., Tang, W., Qin, J., and Cheng, C. C. K.: On downward shortwave and longwave radiations over high altitude regions: Observation and modeling in the Tibetan Plateau, Agric. For. Meteorol, 150, 38-46, 10.1016/j.agrformet.2009.08.004, 2010a.

2.  *It is not clear how to estimate turbidity from AOD at 550nm, my understanding is that Angstrom wavelength exponent over land is not good enough for the extrapolation.*

**Author Response:** Thank you for this comment. In this study, turbidity was estimated via AOD at 550 nm and the Angstrom wavelength exponent, with the Angstrom wavelength index assumed to be 1.3. The Angstrom wavelength exponent varies with aerosol type, aerosol size, and climate conditions. Therefore, the Angstrom wavelength exponent over land is not sufficient for extrapolation. Nevertheless, the method adopted in this study has been used by several studies (Kim, 2004; Yang et al., 2006; Huang et al., 2018), and the derived DSR has also been validated with high accuracy. This proves that the method is suitable for some study areas. Moreover, due to the high altitude of the TP, the air is clean, and the aerosol load is low (Xin et al., 2007; Xia et al., 2008). The uncertainty caused by the Angstrom wavelength exponent is relatively small compared to other regions (Chen et al., 2012; Roupioz et al., 2016). A reliable Angstrom wavelength exponent may further improve the accuracy of the estimated DSR. However, this is still a challenging task, and more detailed work needs to be carried out in the future.

The references are as follows:

Kim, D.-H.: Aerosol optical properties over east Asia determined from ground-based sky radiation measurements, Journal of Geophysical Research, 109, 10.1029/2003jd003387, 2004.

Yang, K., Koike, T., and Ye, B.: Improving estimation of hourly, daily, and monthly solar radiation by importing global data sets, Agric. For. Meteorol, 137, 43-55, 10.1016/j.agrformet.2006.02.001, 2006a.

Huang, G., Liang, S., Lu, N., Ma, M., and Wang, D.: Toward a broadband parameterization scheme for estimating surface solar irradiance: Development and preliminary results on MODIS products, Journal of Geophysical Research: Atmospheres, 123, 12,180-112,193, 10.1029/2018jd028905, 2018.

Xin, J., Wang, Y., Li, Z., Wang, P., Hao, W. M., Nordgren, B. L., Wang, S., Liu, G., Wang, L., Wen, T., Sun, Y., and Hu, B.: Aerosol optical depth (AOD) and Ångström exponent of aerosols observed by the Chinese Sun Hazemeter Network from August 2004 to September 2005, Journal of Geophysical Research, 112, 10.1029/2006jd007075, 2007.

Xia, X., Wang, P., Wang, Y., Li, Z., Xin, J., Liu, J., and Chen, H.: Aerosol optical depth over the Tibetan Plateau and its relation to aerosols over the Taklimakan Desert, Geophysical Research Letters, 35, 10.1029/2008gl034981, 2008.

Chen, Z., Shao, Q., Liu, J., and Wang, J.: Estimating photosynthetic active radiation using MODIS, Journal of Remote Sensing, 16(1), 25-37, 2012.

Roupioz, L., Jia, L., Nerry, F., and Menenti, M.: Estimation of daily solar radiation budget at kilometer resolution over the Tibetan Plateau by integrating MODIS data products and a DEM, Remote Sensing, 8, 10.3390/rs8060504, 2016.

The following reference was added to the revised manuscript:

Kim, D.-H. : Aerosol optical properties over east Asia determined from ground-based sky radiation measurements, Journal of Geophysical Research, 109, 10.1029/2003jd003387, 2004.

3. *It was stated by the authors the accuracy of the retrieval algorithm is highly dependent on the quality of input data, so it seems very necessary to do some sensitivity analysis on the uncertainty produced by potential errors of input products.*

**Author Response:** Thank you for this comment. The accuracy of the parameterization scheme depends on the quality of the input data to some extent. To further understand the effect of uncertainties in input variables on the accuracy of the DSR retrieval scheme, sensitivity analysis of the DSR to input variables is conducted (Fig. 9 and Fig. 10). As shown in Fig. 8, three points located in the west, north central, and southeast of the TP are randomly selected for sensitivity tests. The average of each input variable (including air temperature Tair, air pressure Pair, specific humidity SH, ozone layer thickness, aerosol optical depth AOD, surface albedo, cloud effective

[revised manuscript text omitted]

The above information has been added to the revised manuscript. (P22, L459-P24,

L513)

4. *It is necessary to validate retrievals under clear and cloudy conditions, which is helpful for understanding of the uncertainty sources.*

**Author Response:** Thank you for this comment. The validation results under clear sky and all-sky conditions have been added to the revised manuscript. In addition, the DSR estimation results under all-sky conditions were also validated at ten-day and monthly timescales. Relevant statements have been updated in the revised manuscript. (P11, L249-L253).

'As shown in Fig. 3a and 3b, at the instantaneous scale, the RMSE and R of the 1 km DSR under clear sky are 105.34 W m$^{-2}$ and 0.76, respectively, while those of the 1 km all-sky DSR are 158.19 W m$^{-2}$ and 0.70, respectively. The validation results of this study are not as good as those in other plain areas, where RMSE and R are usually approximately 60 W m$^{-2}$ and 0.9 under clear skies, while those of all-sky conditions are approximately 100 W m$^{-2}$ and 0.9, respectively.' (P11, L249-L253)

[Figure]

**Figure 3.** Validation results for the estimated DSR at (a and b) instantaneous scale, (c) ten-day scale and (d) monthly scale. Scatter plots (a) and (b) show the validation results of instantaneous DSR under clear sky and all-sky conditions, respectively. N indicates the number of points. The legend with different colors denotes twelve stations involved in the validation. The units of RMSE, MB and MAE are W m$^{-2}$.

5. *Measurements of DSR, especially the maintenance, data quality control, etc, should be introduced.*

**Author Response:** Thank you for this comment. Relevant statements have been added in the revised manuscript. (P8, L197-L203).

'Plausible value checks, time consistency checks and internal consistency checks were applied to ensure the accuracy and reliability of the observations. The original sampling data with high frequency were uniformly processed into 30 min and hourly average data by data loggers (e.g., CR3000, CR1000) (Campbell Sci., USA). To retain the observations in their original form as much as possible, no further postprocessing processes are taken, except for replacing outliers with missing values (NaN). Meanwhile, periodic inspection, maintenance and calibration are carried out by professional engineers at all stations.' (P8, L197-L203)

---

## Author Response (AR2)

Dear Reviewer:

We would like to sincerely thank the reviewer for the thoughtful comments and suggestions. All comments and suggestions have been considered carefully and well addressed. For clarity, the referees' comments are listed in black italics, and our responses and changes in the manuscript are shown in blue. We also mention where we made necessary changes in the revised manuscript by indicating page and line numbers in our responses. Please see our responses to your comments and suggestions below.

**Response to Reviewer #3**

*I appreciate that the authors have made great explanations and additional experiments on my concerns, yet I have some minor problems before publication.*

Author Response: We would like to thank Reviewer #3 for your insightful and constructive comments. All your comments and suggestions are very helpful for improving our manuscript. We have carefully considered and addressed all of these comments, and revised our manuscript. Please find our point-by-point response below.

*1. From Table 1 in the Response letter, the slope of all stations is minimal, so there may be limited topographic effects on the stations for evaluation. However, the improvement of Case 2 compared with Case 1 is substantial (Table 5), this is strange, could you give more explanations?*

**Author Response:** Thank you for this comment. Although field instruments are usually setup on a flat land surface, some stations are surrounded by high mountains. For example, QOMS is situated at the bottom of the lower Rongbuk Valley, to the north of Mt. Qomolangma. The spatial distribution of slope and azimuth angle around QOMS station is shown in following Fig. 1. The valley around QOMS has a north–south orientation, with a flat bottom of about 1.5 km width, which corresponds to a small slope. The mountains on both sides of the valley are of 600-900 m in height above the ground, sloping from 25° to 30° (Sun et al., 2018). The surface downward solar radiation is composed of direct radiation, diffuse radiation and reflected radiation. As identified by Chen et al. (2013), the solar diffuse radiation can account for 14% of the solar radiation at the surface of the QOMS station. And for the sunny hillsides, reflected solar radiation can be as high as 100 W m$^{-2}$ sometimes. The hillsides facing the solar incoming direction receive more radiation. The west-facing hillsides which are shaded

by the terrain receive a relative low radiation. Therefore, for stations with complex surrounding terrain, the improvement may be significant. The RMSE of QOMS on the instantaneous scale in Case 1 is 165.81 W m$^{-2}$, and the RMSE on the instantaneous scale in Case 2 is 118.74 W m$^{-2}$. Some stations are not surrounded by high mountains, such as NLGS. The spatial distribution of slope and azimuth angle around NLGS station is shown in following Fig. 2. It can be seen that the terrain around NLGS is relatively smooth, compared to the surrounding area of QOMS station. The improvement of NLGS is minimal. The RMSE of NLGS on the instantaneous scale in Case 1 is 183.56 W m$^{-2}$, and the RMSE on the instantaneous scale in Case 2 is 180.77 W m$^{-2}$.

[Figure]

Figure 1. The DEM (a, unit: m), slope (b, unit: degree, 0-90°) and aspect (c, unit: degree) information around QOMS station.

[Figure]

Figure 2. The DEM (a, unit: m), slope (b, unit: degree, 0-90°) and aspect (c, unit: degree) information around NLGS station.

The related references are as follows:

Sun F., Ma Y., Hu Z., Li M., Tartari G., Salerno F., Gerken T., Bonasoni P., Cristofanelli P., Vuillermoz E. Mechanism of daytime strong winds on the northern slopes of Himalayas, near Mount Everest: observation and simulation. Journal of Applied Meteorology and Climatology, 2018, 57(2): 255-272.

Chen, X., Su, Z., Ma, Y., Yang, K., and Wang, B.: Estimation of surface energy fluxes

under complex terrain of Mt. Qomolangma over the Tibetan Plateau, Hydrology and Earth System Sciences, 17, 2013.

*2. I suggest the authors add some discussions about the challenge of validating DSR products over mountains and the limitations of the current evaluation in the paper. Also discuss the limitations of the current method, e.g., not considering the sky view factor and hard to obtain daily mean DSR estimations. I think the in-depth discussion can greatly improve the impact of this paper and point out future studies. See Ma et al. (2023) Estimation of fine spatial resolution all-sky surface net shortwave radiation over mountainous terrain from Landsat 8 and Sentinel-2 data.*

**Author Response:** Thank you for this comment. Some discussions have been added in the revised manuscript as follows. (P26, L561-L572)

'In this study, topographic effects are coupled in the DSR parameterization scheme by taking shading and terrain reflections into account. Sky view factor is also an important factor for DSR in mountainous areas (Ma et al., 2023). Further improvements may be achieved by introducing the sky view factor into the parameterization scheme. It's still a great challenge to evaluate DSR products over mountainous areas. Currently, it is difficult to do fully evaluations for this complex topography due to lack of in situ measurements on different aspect and slopes over the TP (Yan et al., 2020; Ma et al., 2023). Additionally, the generating of daily shortwave radiation datasets remains a challenge. New-generation geostationary satellites with higher temporal and spectral resolutions, such as FengYun-4 and Himawari-8, have been launched successfully (Bessho et al., 2016; Guo et al., 2017). This provides an opportunity to obtain hourly and daily DSR. Moreover, this allows us to further extend this method to obtain more details of surface radiation components over the TP in the future.' (P26, L561-L572)

The related reference has been added in the revised manuscript as follows (P35, L802; P39, L917):

Ma, Y., He, T., Liang, S., McVicar, T. R., Hao, D., Liu, T., and Jiang, B.: Estimation of fine spatial resolution all-sky surface net shortwave radiation over mountainous terrain from Landsat 8 and Sentinel-2 data, Remote Sensing of Environment, 285, 10.1016/j.rse.2022.113364, 2023.

Yan, G., Chu, Q., Tong, Y., Mu, X., Qi, J., Zhou, Y., Liu, Y., Wang, T., Xie, D., Zhang, W., Yan, K., Chen, S., and Zhou, H.: An operational method for validating the

downward shortwave radiation over rugged terrains, IEEE Transactions on Geoscience and Remote Sensing, 1-18, 10.1109/tgrs.2020.2994384, 2020.

*3. Now I understand the method to make temporal upscaling, yet I suggest the authors explain it in the paper (average by instantaneous values), because readers may misunderstand it as averaging from daily mean values (in the DSR fields, most studies focus on daily and monthly mean values).*

**Author Response:** Thank you for this comment. Relevant statements have been added in the revised manuscript to make it clear as follows. (P12, L266)

'It is upscaled to ten-day and monthly timescales via averaging by instantaneous values.' (P12, L266)